# Reawakening Differentiation Therapy in Acute Myeloid Leukemia: A Comprehensive Review of ATRA-Based Combination Strategies

**DOI:** 10.3390/curroncol33010025

**Published:** 2026-01-02

**Authors:** Shinichiro Takahashi

**Affiliations:** 1Division of Laboratory Medicine, Faculty of Medicine, Tohoku Medical and Pharmaceutical University, 1-15-1 Fukumuro, Miyagino-ku, Sendai 983-8536, Japan; shintakahashi@tohoku-mpu.ac.jp; Tel.: +81-22-290-8889; 2Institute of Molecular Biomembrane and Glycobiology, Tohoku Medical and Pharmaceutical University, 4-4-1 Komatsushima, Aoba-ku, Sendai 981-8558, Japan

**Keywords:** all-trans retinoic acid (ATRA), acute promyelocytic leukemia (APL), acute myeloid leukemia (AML), differentiation therapy, combination therapy, epigenetic modulators, signaling pathways, novel therapeutic strategies

## Abstract

All-trans retinoic acid (ATRA) is an established differentiation therapy for acute promyelocytic leukemia (APL), but its effect in other acute myeloid leukemia (AML) subtypes is limited. This single-author review summarizes current evidence on combination strategies designed to enhance ATRA-induced myeloid differentiation. A PubMed search using the keywords “ATRA,” “myeloid,” and “differentiation inducer or enhancer” identified over 500 published studies as of November 2025. Clinical trials demonstrate that ATRA with arsenic trioxide or epigenetic modulators induces high remission rates in APL and select AML subtypes. Pre-clinical studies reveal synergistic differentiation when ATRA is combined with Cyclin-Dependent Kinase (CDK) and kinase inhibitors, nucleotide synthesis inhibitors, DNA-damaging drugs, B-cell lymphoma 2 (Bcl-2)/Mouse double minute 2 homolog (MDM2) inhibitors, proteasome inhibitors, cytokines, glycosylation modifiers, or natural and antibiotic-derived compounds. Mechanisms include modulation of signaling pathways, chromatin remodeling, RARα stabilization, and interference with nucleotide metabolism. These findings support further clinical investigation of ATRA-based combination therapies beyond APL.

## 1. Introduction

Differentiation therapy by all-trans retinoic acid (ATRA) is a well-established treatment for acute promyelocytic leukemia (APL), a specific type of acute myeloid leukemia (AML). APL is characterized by the promyelocytic leukemia (PML), retinoic acid receptor alpha (RARα) fusion gene, which causes a maturation arrest at the promyelocyte stage, while AML more broadly is a heterogeneous group of myeloid malignancies defined by differentiation blocks at various developmental stages.

ATRA has demonstrated efficacy in APL patients, with few side effects. However, in general, the narrow application of ATRA to APL patients is an obstacle. Therefore, the expansion of differentiation therapies beyond APL and the development of efficient myeloid differentiation enhancers are ongoing avenues of research [1,2]. Moreover, the combination of ATRA with various differentiation enhancers is an attractive strategy for patient treatment.

Recently, a very interesting review has been published [3], which focuses on how the effect of differentiation therapy depends on the status of AML maturation. For example, they demonstrated that the Bcl-2 inhibitor venetoclax and the pan histone deacetylase (HDAC) inhibitor panobinostat exhibited potent activities against less-differentiated and more-differentiated AML states respectively [3,4].

In this review, to gain more efficient and broader differentiation therapy in the future, I summarize the current research on efficient differentiation enhancers combined with ATRA, in a myeloid context generally. These differentiation enhancers include arsenic trioxide (ATO), cyclin-dependent kinase (CDK) inhibitors, kinase inhibitors, epigenetic modifiers, nucleotide biosynthetic pathway inhibitors, DNA damaging agents, B-cell lymphoma 2 (Bcl-2) and Mouse double minute 2 homolog (MDM2) inhibitors, proteasome inhibitors, cytokines, glycosylation modifiers, extracted natural products, especially those that can serve as antibiotics, synthesized small molecule compounds, and repurposed agents with unclear mechanisms of action.

The aim of this review is to provide a comprehensive overview of the current research on ATRA-based combination strategies for myeloid differentiation, highlighting both clinical and pre-clinical studies. By systematically summarizing these differentiation enhancers and their mechanistic basis, this review seeks to offer a resource for researchers and clinicians aiming to expand the applicability of differentiation therapy beyond APL.

A PubMed search from inception to November 2025 using the terms “ATRA,” “myeloid,” and “differentiation inducer or enhancer” identified over 500 articles. Studies were included if they examined ATRA-based combination strategies that promote myeloid differentiation in clinical trials, in vivo models, or mechanistic in vitro systems. Studies on non-myeloid cancers, non-ATRA retinoids, or single-agent ATRA without combination partners were excluded. In addition, reference lists of relevant review articles were manually screened to identify further eligible studies; formal clinical trial registry searches were not performed. Articles meeting these criteria were incorporated into this narrative review.

The principal novelty of this review is its comprehensive and unusually broad coverage of ATRA-based combination strategies. In addition to well-established partners such as arsenic trioxide (ATO) and epigenetic modulators, this manuscript systematically integrates data on a wide spectrum of agents—including CDK and kinase inhibitors, nucleotide synthesis inhibitors, DNA-damaging agents, Bcl-2/MDM2 inhibitors, proteasome inhibitors, cytokines, glycosylation modifiers, natural products, and antibiotic-derived compounds. By compiling clinical findings together with pre-clinical in vivo and mechanistic in vitro studies across this diverse range of therapeutic classes, this review provides one of the most extensive and unified summaries to date of ATRA-enhanced myeloid differentiation.

## 2. Clinical Studies of ATRA-Based Combination Therapies

### 2.1. ATRA and ATO in APL

Among the enhancers described in this chapter, the author emphasizes the strengths not only of ATRA and ATO [5,6,7], but also their combination with epigenetic modifiers [8,9,10,11,12,13,14,15,16,17], since many clinical studies of ATO and epigenetic modifiers have been published so far.

Clinical studies of the combination of ATRA and ATO have been reported [5,6,7] (Table 1), and the combination of these agents is considered standard therapy for APL. A recent study showed that oral arsenic is highly effective in high-risk APL [5]. The authors evaluated oral arsenic and ATRA without chemotherapy as an outpatient consolidation therapy in 54 high risk APL patients. Complete molecular remission was achieved at the end of the consolidation phase in all patients. Tang et al. [6] reported a retrospective study of 212 non-high-risk APL patients who received ATRA plus oral arsenic or intravenous ATO. The 5-year overall survival, event-free survival, and cumulative incidence of relapse were, 96.3%, 92.3%, and 5.5%, respectively. The authors also found that PML-RARα transcript levels at the end of induction therapy were associated with prognosis in their cohort. Another group evaluated oral arsenic or intravenous ATO in phase III study. After a median follow-up of 32 months, 34 (94%) of 36 patients in the ATO-ATRA group, and 67 (97%) of 69 patients in the oral arsenic-ATRA group, achieved 2-year event-free survival [7]. The strengths of the combination with ATO lie in these promising results from clinical studies.

### 2.2. ATRA Combined with Epigenetic Modifiers

In addition to ATO, epigenetic modifiers are also promising agents as differentiation enhancers (Table 1). Cimino et al. [8] performed a pilot study of eight refractory or high-risk AML patients, not eligible for intensive therapy, treated with VPA followed by the addition of ATRA. In two cases, hematologic improvement was observed. Stable disease was observed in five cases and disease progression was observed in one case. The authors concluded that administration of VPA, followed by ATRA is well tolerated and induces phenotypic changes of AML blasts. Another group reported a clinical trial of VPA and ATRA in 26 patients with poor-risk AML (7 patients later withdrew from the trial). The authors found that treatment with VPA and ATRA resulted in transient disease control in a subset of patients with AML that progressed from a myeloproliferative disorder (n = 2) but not in patients with primary (n = 11) or myelodysplastic syndrome (MDS)-related (n = 6) AML [9]. In a phase II study of 75 patients with MDS and AML, Kuendgen et al. [10] concluded that VPA is clinically useful in low-risk MDS. They concluded that VPA may be combined with demethylating drugs or chemotherapy for patients with high-risk MDS. In addition, ATRA has the potential to induce a prolonged second response in patients with high-risk MDS. Raffoux et al. [11] examined the effect of VPA and ATRA in 11 elderly patients with de novo AML (median age 82); the authors observed complete marrow response in 3 patients, including 1 patient with CR, and hematologic improvement in 2 additional patients [11]. Thomas et al. [12] reported that, in a small cohort of 5 elderly AML patients, a novel biomodulatory therapy with low-dose 5-azacitidine (AZA) combined with PPARγ ligands (e.g., pioglitazone [PGZ]), and ATRA (APA) induced complete molecular remission in primary chemorefractory disease. Three out of five patients achieved CR with residual thrombocytopenia, strong myeloid differentiation in leukemia blasts, and residual haematopoiesis during APA treatment. The authors concluded that combined biomodulation with APA may act synergistically on leukemic differentiation and growth control. They further investigated these combinations in vitro and found that treatment with ATRA/AZA/PGZ enhanced phagocytic activity and increased production of reactive oxygen species in AML blasts [19]. In a randomized phase II clinical study, it was reported that the addition of ATRA, but not VPA, to decitabine improved clinical outcomes [13]. In patients with difficult-to-treat disease, the addition of ATRA to decitabine resulted in a clinically meaningful extension of survival and a higher remission rate, without added toxicity [13]. Another group evaluated the efficacy of the combination of AZA, VPA, and ATRA in patients with high-risk AML or MDS and reported a 26% response rate, including 22% CR rate, with AZA/VPA/ATRA treatment in patients with high-risk AML or MDS. The authors concluded that in older patients with AML, direct randomized comparisons to other conventional care regimens should be examined for these epigenetic modulation regimens, at least in those with a relatively low blast count with previously untreated disease [14].

Clinical evidence is still preliminary, with small and heterogeneous cohorts, modest responses in non-APL AML, and few randomized trials. Larger studies are needed to confirm clinical benefit.

Collectively, these studies show the strengths of ATRA-based combinations with ATO or epigenetic modifiers, as multiple clinical investigations have reported encouraging findings.

## 3. Pre-Clinical Strategies Enhancing ATRA-Induced Differentiation

### 3.1. Combination of ATRA with CDK Inhibitors

The CDK4/6/cyclin D pathway is a key initiator of the G1-S phase transition, which determines cell fate. Reports of CDK inhibitors for myeloid differentiation are relatively scarce, and only five pre-clinical studies have been published to date (Table 2). Hu et al. [20] recently investigated the effects of the CDK4/6 inhibitor palbociclib with ATRA on promoting leukemia differentiation in vitro and in vivo. The authors demonstrated that palbociclib sensitized AML cells to ATRA-induced morphologic and functional changes that indicate myeloid differentiation. Novikova et al., revealed that CDK6 and several other genes are the key regulators of the molecular responses to the ATRA treatment in leukemia cell lines (HL-60, NB4 and K562), and claimed that the combination of palbociclib and ATRA could be effective [21]. Roscovitine is a purine analogue that inhibits the activity of CDKs by targeting their ATP-binding pockets [22]. Roscovitine enhanced the nuclear enrichment of specific traditionally cytoplasmic signaling molecules, like Src-family kinase members, and enhanced the differentiation and cell cycle arrest of HL-60 AML cells [23,24]. Sao et al. [25] demonstrated that both CDK2 depletion and the pharmacological inhibitor of CDK2 (by SU9516) significantly sensitized three subtypes of AML cells, U937, HL-60, and NB4 cells, to ATRA-induced cell differentiation. A Phase I clinical trial has been conducted to study optimal dosing of palbociclib alone, or in combination with either dexamethasone, decitabine or sorafenib in advanced AML (NCT03132454). From this study, although well tolerated, palbociclib showed minimal activity. Several other studies, including phase Ib (NCT03878524) and phase I/II (NCT03844997) trials are ongoing, but results are not available yet. Therefore, the clinical translation of CDK inhibitors as differentiation enhancers remains limited at present.

### 3.2. Combination of ATRA with Kinase Inhibitors

The author recently reviewed therapeutic strategies in AML, highlighting kinase pathways such as FLT3, PI3K/AKT/mTOR, CDK, and CHK1, and emphasizing the potential of combination therapies over monotherapies [50]. As mostly covered in that review, more than 20 papers have reported pre-clinical findings on the effects of kinase inhibitors on enhancing myeloid differentiation (Table 2). Detailed mechanistic schematics of these pathways have been presented in my previous reviews [1,2]. ATRA induces activation of p90-kDa ribosomal S6 kinase (p90RSK) and inactivation of glycogen synthase kinase 3β (GSK3β), which increases myeloid cell leukemia-1 (Mcl-1) levels [26]. Sorafenib, a multikinase inhibitor, was shown to block the ATRA-induced Mcl-1 increase by reversing p90RSK activation and GSK3β inactivation, and to augment apoptosis and induction of differentiation in a non-APL AML cell line and primary cells [26]. Src family kinases (SFKs) are hyperactivated in AML [51]. SFKs inhibit the activity of the retinoic acid receptor alpha (RARα [27], and SFK inhibitors (PP2, dasatinib) enhanced ATRA-mediated cellular differentiation in an AML cell line and primary blasts via Lyn inhibition-mediated activation of RAF-1/MEK/ERK [28,29,30,31]. The SFK inhibitor PP2 in combination with ATRA and arsenic trioxide induced APL cell differentiation [32]. Radotinib, a BCR/ABL tyrosine kinase inhibitor, promoted differentiation and induced CD11b expression by downregulating Lyn in AML cells [33]. Redner et al. [18] performed a phase-I dose escalation study of the combination of ATRA and the SFK inhibitor dasatinib in patients with high-risk myeloid neoplasms. However, no clinical responses were observed (Table 2). In vitro studies showed that autophagy contributes to dasatinib-induced differentiation of AML cells [34]. Enzastaurin, a derivative of the protein kinase C (PKC) pan-inhibitor staurosporine, suppresses the activation of PKCβ [52]. The combination of enzastaurin and ATRA exerted dose-dependent effects on ATRA-resistant APL cells for inducing apoptosis [35] and showed an efficient effect on ATRA induced differentiation in AML cells [36]. Similarly, staurosporine combined with ATRA exerted synergistic effects in promoting myeloid differentiation in monocytic leukemia U937 cells with low sensitivity to ATRA, but not in erythroleukemia K562 and AML Kasumi cells unresponsive to ATRA [37]. This induction was mediated by MEK/ERK activation and modulation of the protein level of CCAAT/enhancer-binding protein beta (C/EBPβand C/EBPε in U937 cells. Zhang et al. [38] reported that the combination of ATRA with salt-inducible kinase (SIK) inhibition significantly enhanced ATRA-mediated AML differentiation. The combination of ATRA and SIK inhibition synergistically activated Akt but not the MAPK signaling pathway, and Akt inhibition abrogated the effect of SIK inhibition. In contrast, Wang et al. [39] recently reported that the sequential treatment of ATRA plus PI3K/Akt inhibitors was as an efficient strategy for AML therapy. The authors found that genes translationally regulated by ATRA were mainly enriched in PI3K/Akt signaling. Therefore, the mechanisms of Akt and SIK inhibitors in enhancing the effect of ATRA are different. In addition, the mammalian target of rapamycin (mTOR) is a downstream target of PI3K/Akt, and inhibition of mTOR by its inhibitor RAD001 (everolimus) potentiated the effects of ATRA to induce growth arrest and differentiation of human AML cell lines NB4 and HL-60 [40]. Total and phosphorylated epidermal growth factor receptor proteins were increased in approximately 15% of patients with AML compared with healthy CD34+ samples [53]. De Almeida et al. [41] reported that the combination of gefitinib with ATRA and ATO rewired NB4-R2 (ATRA-resistant) and NB4 ATOr (ATO-resistant) cells to acquire sensitivity to standard therapy for APL. APL mice treated with ATRA alone or in combination with gefitinib exhibited increased overall survival in comparison with the vehicle-treated group [41]. Lainey et al. [42] demonstrated that EGFR inhibitors, gefitinib and erlotinib alone did not promote differentiation but stimulated the acquisition of biochemical maturation markers and morphological change when combined with either ATRA or vitamin D. Lu et al. [43] demonstrated that trametinib, a highly selective inhibitor of MEK, enhanced ATRA-induced differentiation in AML cell lines U937, HL-60 and primary AML cells. Moreover, this combination restored ATRA sensitivity in the ATRA-resistant AML cell line HL-60Res. FLT3 is a receptor tyrosine kinase and one of the most frequently mutated genes in AML, and the commonest mutation of FLT3 is internal tandem duplication (ITD). Mutation or over-expression of FLT3 results in the constitutive activation of downstream pathways and aberrant cell growth [54]. Ma et al. [44] reported the synergistic activity of the ATRA and FLT3 inhibitors (Sorafenib, AC220, TTT-3002) in inducing apoptosis in FLT3-ITD positive primary patient samples and cell lines. Selectively targeting Janus kinase (JAK) 1 has been explored as a means to suppress inflammation and signal transducers and activators of transcription (STAT)-associated pathologies related to proliferation, immune response, and malignancy [55]. Ramsey et al. [45] demonstrated that selective inhibition of JAK1 primes STAT5-driven human leukemia cells for ATRA-induced differentiation. The authors demonstrated the synergistic activity of JAK1 inhibition and ATRA in non-APL leukemia. Shao et al. [46] demonstrated that TAK165, a human epidermal growth factor receptor 2 inhibitor, exhibited potent synergy with ATRA to promote differentiation of AML cells. The enhanced differentiation was dependent on the activation of the RARα/STAT1 axis, and they further revealed that the MEK/ERK cascade regulates the activation of STAT1. Although a number of in vitro studies of kinase inhibitors have been published, weaknesses of the studied combinations remains, and they have shown very limited effects [18].

### 3.3. Combination of ATRA with ATO (Pre-Clinical Studies)

Based on the results of clinical studies of ATRA with ATO [5,6,7], this combination is now employed as a standard therapy for APL. Consequently, the mechanisms of this combination have been studied through in vitro experiments. These are shown in Table 2. Co-treatment with ATO enhances ATRA-induced HL-60 differentiation by altering the expression of genes involved in cell differentiation [47]. ATO induces the rapid Small ubiquitin-like modifier (SUMO)ylation, the attachment of a small ubiquitin-related modifier to a target protein, of the PML/RARα oncoprotein, resulting in its elimination by the ubiquitin/proteasome system [56]. Inhibition of the SUMO pathway by chemotherapy-induced reactive oxygen species is required for the rapid and essential cell death of chemosensitive non-APL AML cells subjected to anthracyclines or Ara-C treatment [57]. Baik et al. [48] demonstrated that targeting the SUMO pathway by pharmacologic inhibitors such as 2-D08 or anacardic acid restored the prodifferentiation and antiproliferative activities of retinoids in non-APL AML. ATO cooperates with ATRA to enhance MAPK activation and differentiation in APL-RARα-negative AML cells. ATO enhanced ATRA-induced RAF/MEK/ERK signaling, expression of CD11b and p47(PHOX), and inducible oxidative metabolism [49].

### 3.4. Combination of ATRA with Epigenetic Modifiers (Pre-Clinical Studies)

Not only clinical, but also pre-clinical reports have been published on epigenetic modifiers (Table 3). In 2014, enhanced differentiation by ATRA and histone deacetylase inhibitor valproic acid (VPA) was shown using APL NB4 cells [58]. Thereafter, the combination of VPA and ATRA promoted differentiation in ATRA-sensitive and -resistant cell lines, depending on the induction of autophagic flux [59]. Moretti et al. [60] demonstrated that VPA differentiated the bulk APL cells and vorinostat (or suberoylanilide hydroxamic acid, SAHA) selectively targeted leukemia-initiating cells (LICs) in a murine model. The ATRA + VPA + SAHA combination efficiently induced complete remission (CR) in an APL model. This combination also lowered LIC levels. A previous study using an APL mouse model showed that VPA induced rapid tumour regression and extended survival [61]. However, termination of VPA led to immediate relapse. The authors concluded that VPA spares leukemia-initiating activity in mouse models of AML [61].

Previous studies demonstrated that troglitazone, a ligand for peroxisome proliferator activated receptor gamma (PPARγ), combined with retinoid was a potent inhibitor of growth in several myeloid cell lines [62].

A pre-clinical study of the HDAC class I selective inhibitor entinostat was reported [63]. Using AML cell line Kasumi-1, HL-60 and primary AML blasts, they concluded that entinostat is at least as effective as decitabine. Trichostatin A induced differentiation in erythroid cell lines and synergistically induced the differentiation by ATRA or vitamin D3 in several myeloid (U937, NB4, and HL-60) cells, and also ATRA resistant NB4 and HL-60 cells [64]. Sodium phenylbutyrate (SB) is a well-known HDAC inhibitor; it is difficult to obtain therapeutic serum concentration above effective levels. Yu et al. [65] demonstrated that the combination of SB with ATRA resulted in an improved differentiation effect with clinically achievable levels of SB.

Approximately 15% of AML patients harbor mutations in isocitrate dehydrogenase (IDH), which leads to the production of the oncometabolite 2-hydroxyglutarate (2-HG) [66]. Inhibitors of mutant IDH enzymes, which suppress 2-HG production, have been approved by the FDA for use in patients [67]. In IDH1 mutated AML cells, ATRA effects were significantly reduced by the inhibition of 2-HG production, whereas in wild-type IDH1 cells, the treatment with a cell-permeable form of 2-HG sensitized cells to ATRA-induced myeloid differentiation [68]. Vitamin D enhanced ATRA effects in IDH1 mutated AML patients through the 2-HG/C/EBPα/vitamin D receptor axis [69]. A previous study using tranylcypromine (TCP), an inhibitor of the histone H3 lysine 4 (H3K4) demethylase lysine-specific demethylase (LSD) 1, demonstrated that drug-induced epigenetic remodeling reprogrammed AML cells to respond to ATRA (tretinoin)-based therapy [70]. A similar result was reported by Barth et al. [71]. Following these results, clinical trials using TCP and a number of LSD1 inhibitors are now underway [15,16,17].

Hyaluronic acid is a major component of the extracellular matrix that selectively binds the CD44 transmembrane receptor that is overexpressed in most primary cancers [72]. Butyric acid is the smallest HDAC inhibitor. It was previously demonstrated that a novel retinoic/butyric hyaluronan ester may be useful in controlling the differentiation of RA-sensitive cells and the proliferation of RA-resistant cells [73].

Selenite is a redox active selenium compound that is implicated in the removal of zinc from zinc/thiolate coordination sites [74]. Selenite was reported to inhibit the activity of DNA methyltransferase (DNMT) [75]. Selenite is part of a novel class of cancer chemotherapeutic agents with superior cytotoxic effects on many cancer cells [76]. Misra et al. [77] reported that selenite, at a clinically achievable dose, targets PML/RARα for degradation. The authors showed that, in combination with ATRA, selenite potentiates the differentiation of APL cells.

**Table 3 curroncol-33-00025-t003:** Pre-clinical studies of ATRA with epigenetic modifiers and DNA/nucleotide synthesis inhibitors. Pre-clinical indicates studies limited to experimental models.

**Pre-Clinical Studies of Epigenetic Modifiers**			
**Differentiation Agent(s)**	**Action**	**Model Level**	**Ref**
Valproic acid (VPA) (HDAC inhibitor)	VPA suppressed NB4 cell proliferation, an effect that was potentiated by ATRA. Co-treatment also upregulated myeloid transcription factors (C/EBPα, β, ε, and PU.1), facilitating differentiation.	Cell line (NB4 cells).	[58]
VPA	VPA combined with ATRA promoted autophagy and differentiation in ATRA-sensitive NB4 cells and also in ATRA-resistant NB4R and THP-1 cell lines.	Cell line (NB4, ATRA-resistant NB4R and THP-1 cells).	[59]
VPA, vorinostat/suberoylanilide hydroxamic acid (SAHA) (HDAC inhibitor)	In an APL mouse model, SAHA was shown to target leukemia-initiating cells. Co-treatment with ATRA, VPA, and SAHA effectively induced complete remission and decreased LIC frequency.	In vivo (APL model mice).	[60]
VPA	In several APL mouse models, VPA induced terminal differentiation; however, discontinuation of VPA led to rapid relapse. Moreover, VPA increased LIC activity. Unlike ATRA or arsenic, VPA did not promote degradation of PML-RARA.	Primary blasts/in vivo (PML-RARA-transformed primary hematopoietic progenitors and APL mouse models).	[61]
Troglitazone (an antidiabetic drug, also identified as a ligand for PPAR gamma).	Co-treatment with troglitazone and a ligand selective for RAR (ATRA, ALART1550), RXR (LG100268), or both receptors (9-cis RA) effectively inhibited clonal growth in several myeloid leukemia cell lines.	Cell line (NB4, HL-60, U937, ML-1 and THP-1 cells).	[62]
Low-dose AZA combined with PPARγ ligands [e.g., pioglitazone (PGZ)], and ATRA	In HL-60 and U937 cells, as well as in about 50% of primary AML samples, the drug combination effectively suppressed proliferation and promoted differentiation. AML blasts treated with ATRA, AZA, and PGZ exhibited increased ROS levels and phagocytic activity.	Cell line/primary blasts (HL-60, MV4-11, MOLM-13, U937 cells, 14 primary AML cells).	[19]
Entinostat (HDAC class-I selective inhibitor)	Entinostat induced differentiation in AML cell lines and primary AML cells, with this effect being further augmented by ATRA. Acting as a priming agent for ATRA-mediated differentiation, entinostat exerts its effects independently of RARβ2.	Cell line/primary blasts (Kasumi-1, HL-60, NB-4, U937, K562, KG-1 and 46 primary AML blasts).	[63]
Trichostatin A (TSA), trapoxin A (TPX) (HDAC inhibitors [HDIs])	TSA and/or TPX induced differentiation in both myeloid (e.g., U937) and erythroid (e.g., K562) cell lines. Co-treatment with ATRA resulted in a synergistic enhancement of differentiation. In clinical AML specimens ranging from M0 to M7, TSA alone elicited morphological and phenotypic changes in 12 of 35 samples (34%).	Cell line/primary blasts (K562, HEL, U937, HL60, HL60/RA (ATRA resistant HL60), NB4, MEG-O1 cells and 35 clinical specimens from AML).	[64]
Sodium phenylbutyrate (SB)(HDAC inhibitor)	SB in combination with ATRA synergistically inhibited colony formation and promoted CD11b expression. The combination significantly affected S-phase progression, with the interaction shifting from antagonistic at low ATRA concentrations to synergistic at higher levels (>0.5 µM).	Cell line (ML-1 cells).	[65]
Cell-permeable form of 2-hydroxyglutarate (2-HG)	AML blasts with IDH1 mutations generate 2-HG, leading to hypermethylation. ATRA selectively impaired viability and induced apoptosis in these cells. Cell-permeable 2-HG sensitized wild-type AML cells to ATRA-induced differentiation. In vivo, ATRA reduced tumor burden and prolonged survival in mice bearing mutant IDH1 AML.	Cell line/primary blasts/in vivo (HL-60, MOLM14, NB4, 14 primary AML patient samples. A xenograft model based on immunodeficient NOD–scid IL2rγnull (NSG) mice with primary AML samples, or MOLM14 carrying the IDH1– R132H mutation).	[68]
2-HG	2-HG specifically activates the vitamin D receptor (VDR) in IDH-mutant AML cells, increasing their sensitivity to the combination of ATRA and vitamin D (or a VDR agonist).	Cell line/primary blasts/in vivo (HL60, U937, KG1a, THP1IDH1WT, THP1IDH1R132H, HL60IDH2WT, HL60IDH2R172K, 24 primary AML patient samples, a xenograft model).	[69]
Tranylcypromine (TCP) (Lysine-Specific Demethylase 1 (LSD1) Inhibitor)	Inhibition of LSD1 enhanced H3K4 dimethylation, especially at myeloid differentiation-related genes. TCP combined with ATRA significantly suppressed engraftment of primary human AML cells in NOD-SCID mice and showed stronger anti-leukemic activity, targeting leukemia-initiating cells, than either treatment alone.	Cell line/primary blasts/in vivo (HL-60, TEX [derived from primitive human cord blood cells, ATRA insensitive] cells. Normal bone marrow mononuclear cells, primary AML cells (n = 5), umbilical cord blood cells (n = 5). In vivo treatment of AML in NOD-SCID and NSG mice, Secondary transplants of AML-engrafted mice).	[70]
A novel retinoic/butyric hyaluronan ester (HBR)	In RA-sensitive NB4 cells, HBR promoted terminal differentiation and growth arrest, while in RA-resistant NB4.007/6 cells, it inhibited proliferation through apoptosis. Treatment with HBR significantly increased survival in NB4- or P388-xenografted mice.	Cell line/in vivo (NB4, and on its RA-resistant subclone, NB4.007/6, SCID/NB4 model and the P388 lymphocytic leukemia in DBA mice).	[73]
Selenite (DNMT inhibitor)	By targeting PML/RARα for degradation, selenite suppressed survival and proliferation of NB4 cells. While selenite alone did not induce differentiation, it potentiated ATRA-mediated differentiation in these cells.	Cell line (NB4).	[77]
**Pre-Clinical Studies of De Novo Nucleotide Biosynthetic Pathway Inhibitors and DNA Damaging Agents**			
**Differentiation Agent(s)**	**Action**	**Model Level**	**Ref**
ML390, BRQ (dihydroorotate dehydrogenase (DHODH)inhibitors)	In ER-homeobox (HOX) A9–transduced primary murine bone marrow cells, terminal differentiation occurs following β-estradiol withdrawal. Through a phenotypic screen using this model, DHODH inhibitors were found to bypass the differentiation block, reduce leukemia-initiating cells, decrease leukemic burden, and enhance survival.	Cell line/primary blasts/in vivo (THP-1, U937, ER-HoxA9 GMP Cell Lines, the HoxA9 + Meis1 or MLL/AF9 primary leukemia cells. Subcutaneous xenograft tumor mice models, disseminated intravenous xenograft leukemia mice models, patient AML sample engrafted (PDX) mice).	[78]
BAY 2402234 (DHODH inhibitor)	BAY2402234 induces differentiation in many myeloid cell lines, and AML cell line xenografts, as well as PDX model.	Cell line/in vivo (THP-1, MV4-11, TF-1, MOLM-13, HEL, SKM-1, NOMO-1, UOC-M1 and EOL-1 cells. Tumor xenografted NOG or NOD/SCID mice).	[79]
5-Aminoimidazole-4-carboxamide ribonucleoside (AICAr)	AICAr enhanced ATRA-driven differentiation in NB4 cells and independently induced monocyte–macrophage markers in U937 cells, effects that were mediated via MAPK activation.	Cell line (HL-60, NB4, U937)	[80]
AICAr, brequinar (DHODH inhibitor)	AICAr induced macrophage-like differentiation in a subset of primary non-APL AML blasts. RNA-seq analysis demonstrated that this treatment inhibited pyrimidine metabolism.	primary blasts (35 primary AML cells)	[81]
Triciribine (Akt inhibitor and inhibitor of nucleotide synthesis)	In NB4 and HL-60 cells, differentiation correlated with ERK activation. Triciribine treatment enriched pathways related to cytokine–cytokine receptor interactions and hematopoietic cell lineage, according to pathway analysis.	Cell line (NB4, HL-60 cells).	[82]
6-benzylthioinosine (6BT), a closely related compound of 6-methylthioinosine, which is a potent inhibitor of de novo purine synthesis	6BT induced monocytic differentiation and cell death in myeloid leukemia cell lines, with minimal cytotoxicity toward nonmalignant cells, including fibroblasts, normal bone marrow, and endothelial cells. In xenografted mice, 6BT effectively inhibited the growth of MV4-11 and HL-60 tumors.	Cell line/primary blasts/in vivo (HL-60, OCI-AML3, OCIM2, MV-411, HNT34 cells. 5 primary AML samples. fibroblasts, normal bone marrow, and endothelial cells. HL-60 or MV-411 xenograft mice).	[83]
Pyrimethamine (PMT)(dihydrofolate reductase [DHFR] antagonist)	Oral PMT treatment was effective in two xenograft mouse models. PMT strongly inhibited human AML cell lines and primary patient cells, while sparing CD34+ hematopoietic cells from healthy donors.	Cell line/primary blasts/in vivo (Human AML cell lines, primary patient cells, two xenograft mice models, and CD34+ cells from healthy donors).	[84]
Topotecan (TPT)(topoisomerase I inhibitor)	TPT synergized with ATRA to induce DNA damage and trigger caspase-dependent apoptosis, with RARα mediating this effect. The combined efficacy was confirmed in HL-60 xenografted mice.	Cell line/in vivo (HL60, NB4, U937 cells and HL60 xenografted nude mice).	[85]
Aclacinomycin (ACLA) (topoisomerase I/II inhibitor)	ATRA and ACLA induced granulocytic differentiation in HL-60 and NB4 cells, concomitant with increased migratory and invasive activity. ACLA-driven differentiation upregulated MMP-9, whereas ATRA decreased MMP-9 and induced urokinase plasminogen activator mRNA expression.	Cell line (HL-60, NB4 cells).	[86]
ICRF-154, 193 (topoisomerase II inhibitor)	Both ICRF-154 and ICRF-193 promoted differentiation of APL cell lines and primary cells from APL patients, and synergized with ATRA to suppress cell proliferation and enhance differentiation.	Cell line/primary blasts (NB4, HT-93, HL-60, U937 and 3 primary APL cells).	[87]
1-(2-deoxy-2-methylene-beta-D-erythro-pentofuranosyl) cytidine (DMDC)(cytidine deaminase-resistant analogue of ara-C)	DMDC suppressed proliferation of APL and AML cell lines and promoted differentiation in APL cells. In NB4 cells, DMDC combined with ATRA induced differentiation synergistically, with comparable effects observed in primary APL patient cells.	Cell line/primary blasts (HL-60, NB4, U937, and HT93 cell line. 3 primary APL cells).	[88]
Ara-C (pyrimidine nucleoside analog)	Both ATRA and ara-C triggered apoptosis in CML cells, with ara-C showing greater efficacy. Their combined treatment resulted in an additive, rather than synergistic, effect.	Primary blasts (Freshly isolated cells from 10 patients with chronic-phase CML).	[89]
2′-deoxycoformycin (dCF), 9-beta-D-arabinofuranosyladenine (Ara A), fludarabine (FLU), cladribine (CdA)(deoxyadenosine analogs)	Combined treatment with dCF and Ara A effectively induced differentiation in monocytoid leukemia cells (U937, THP-1, P39/TSU, JOSK-M). Among myeloid leukemia cells (NB4, HL-60), CdA was the most potent analog in promoting differentiation, with or without ATRA.	Cell line (K562, HL-60, NB4, KG-1, ML-1, U937, THP-1, P39/TSU, JOSK-M cells).	[90]
dCF and 2′-deoxyadenosine (dAd) (adenosine deaminase inhibitor)	NB4 cells exhibited granulocytic differentiation in response to ATRA or dAd plus dCF, but not to ara-C. Pre-treatment with ATRA enhanced the differentiation effect of dAd plus dCF, whereas pretreatment with dAd plus dCF before ATRA had a reduced impact.	Cell line (K562, HL-60 and NB4 cells).	[91]
Neplanocin A (NPA, a potent S-adenosylhomocysteine hydrolase inhibitor), dCF, deoxyadenosine (dAd).	Both NPA and dAdo plus dCF synergized with ATRA to promote myeloid differentiation in NB4 cells. Pre-treatment with ATRA markedly potentiated the differentiation-inducing effect of dAdo plus dCF, whereas pretreatment with dAdo plus dCF prior to ATRA was less effective.	Cell line (NB4, K562, U937 cells).	[92]

### 3.5. Combination of ATRA with De Novo Nucleotide Biosynthetic Pathway Inhibitors and DNA Damaging Agents

Many reports have published pre-clinical findings of the effects of de novo nucleotide biosynthetic pathway inhibitors and DNA damaging agents on myeloid differentiation (Table 3). Cancer cells extensively utilize de novo pathways for nucleotide biosynthesis, since pyrimidines and purines are essential components of DNA synthesis [93]. As enzymes involved in the folate cycle and de novo pyrimidine and purine biosynthesis are among the most highly expressed proteins in cancers, they are promising targets for inhibitors [93].

A previous study showed that inhibition of dihydroorotate dehydrogenase (DHODH), a key enzyme in the de novo pyrimidine synthesis pathway, induces myeloid differentiation in human and mouse AML models [78]. Christian et al. [79] showed that BAY 2402234, an orally bioavailable, potent, selective, and a novel DHODH inhibitor, induces differentiation and exhibits monotherapy efficacy across multiple AML subtypes. By inhibiting UMP-synthase, aminoimidazole-4-carboxamide ribonucleoside (AICAr) interferes with pyrimidine synthesis, a step downstream of DHODH. DHODH inhibitor and AICAr had similar effects on differentiation and S-phase arrest [94]. Consistent with this, another report demonstrated that AICAr induces differentiation of AML cells [80]. AICAr induced differentiation in a subset of primary non-APL blasts, and these effects correlated with sensitivity to potent inhibitors of DHODH, such as brequinar [81].

Many agents that target enzymes in the purine nucleotide biosynthesis pathway have been developed [95,96]. We recently revealed that triciribine (TCN) efficiently induces the differentiation of myeloid cells [82]. TCN functions as an Akt inhibitor [97] and inhibition of Akt was reported to enhance the effect of ATRA [39]. However, the authors suggested that the differentiation effect of TCN may partly depend on the inhibition of nucleotide synthesis, as TCN was reported to inhibit de novo purine nucleotide synthesis [97,98]. From compound library screening, 6-benzylthioinosine (6BT) was identified as a promising differentiation-inducing agent [83]. The closely related compound 6-methylthioinosine is a potent inhibitor of the enzyme amidophosphoribosyltransferase, which catalyses the first committed step in de novo purine synthesis [99]. One study showed that 6BT induces monocytic differentiation of primary AML patient samples and myeloid leukemia cell lines such as OCI-AML3 and HL-60 [83]. Dihydrofolate reductase (DHFR) is an enzyme that catalyses tetrahydrofolate regeneration, using NADPH as an electron donor, by reduction of dihydrofolate [100]. Tetrahydrofolate is required for de novo synthesis of thymidylate, purines, methionine, glycine, and serine [100]. A previous report showed that pyrimethamine (PMT), a DHFR antagonist, potently enhances apoptosis and differentiation in a leukemia cell line identified by high-throughput drug screening [84]. Notably, PMT targets leukemic cells, but not CD34+ cells from healthy donors. The effect of PMT was rescued by excess folic acid, suggesting an oncogenic function of folate metabolism in AML.

Topotecan (10-hydroxy-9-dimethylaminomethyl-(S)-camptothecin, TPT) is derived from camptothecin, a semisynthetic topoisomerase 1 inhibitor. TPT forms a covalent complex between DNA and topoisomerase 1, resulting in DNA damage during transcription and cell replication, finally leading to apoptosis [101]. Xu et al. [85] demonstrated that the combination of TPT and ATRA synergistically increases DNA damage in AML cells and enhances antitumor efficacy in the HL-60 xenograft mice.

Aclacinomycin was reported to promote HL-60 and NB4 cell differentiation. The mechanism of this differentiation is different from that induced by ATRA and involves matrix metalloproteases and urokinase plasminogen activator expression [86]. Nittsu et al. [87] demonstrated the effects of ICRF-154 and ICRF-193, DNA topoisomerase II inhibitors, and several anticancer drugs on the growth and differentiation of the AML (HL-60, U937) and APL (NB4, HT-93) cell lines. The DNA topoisomerase II inhibitors, in combination with ATRA, significantly induced differentiation of APL cell lines and primary APL cells from three cases.

The cytidine deaminase-resistant analogue of araC 1-(2-deoxy-2-methylene-beta-D-erythro-pentofuranosyl) cytidine (DMDC) was reported to inhibit the growth of NB4 and HT-93 cells and was also effective against HL-60 and U937 cells. DMDC was most effective in NB4 cells for inducing differentiation and inhibiting growth. Differentiation of NB4 cells by ATRA was not augmented by araC, whereas combined treatment with ATRA and DMDC produced greater-than-additive effects [88]. Consistent with this finding, ATRA and araC combination induced apoptosis, but not differentiation, in primary CML cells [89].

A previous study showed that monocytoid cells U937 and THP-1 cells are very sensitive to combined treatment with 9-beta-D-arabinofuranosyladenine (Ara A) and 2′-deoxycoformycin (dCF) [90]. Adenosine deaminase-resistant analogues, such as cladribine and fludarabine, also induce differentiation of human myeloid leukemia HL-60 cells or monoblastic leukemia U937 cells. Another study by the same group demonstrated that dCF plus deoxyadenosine induced the differentiation of NB4 cells in combination with ATRA [91]. One report demonstrated that several adenosine analogues, such as neplanocin A, a potent S-adenosylhomocysteine hydrolase inhibitor, and deoxyadenosine plus dCF, in combination with ATRA, greatly enhance granulocytic differentiation of NB4 cells [92]. These various in vitro data highlight the strengths of the combining de novo nucleotide biosynthetic pathway inhibitors and DNA damaging agents, although a major limitation is the lack of clinical studies.

### 3.6. Combination of ATRA with Bcl-2 Inhibitors and MDM2 Inhibitors

The p53 tumour suppressor downregulates antiapoptotic proteins including Bcl-2 and Mcl-1, and upregulates multiple proapoptotic factors, such as Bax and PUMA [102]. Murine double minute 2 (MDM2) is the main negative regulator of p53 [102]. Many Bcl-2 and MDM2 inhibitors have been developed. ABT-737 is a small molecule drug that inhibits Bcl-2 and Bcl-xL, two members of the evolutionarily conserved Bcl-2 family of proteins, sharing Bcl-2 homology (BH) domains [103]. Buschner et al. [104] reported that ABT-737 sensitizes AML-193 and NB-4 cells to ATRA-induced differentiation, along with a reduced uptake of ^18^F-fluorodeoxyglucose (FDG), which is widely used for the diagnosis, staging, and response assessment of malignancies. Thus, ^18^F-FDG uptake could be predictive of the sensitivity to ABT-737. The authors concluded that the combined treatment of ABT-737 and ATRA might represent a promising therapeutic strategy.

Another group demonstrated that JY-1-106, the novel BH3 domain mimetic, which antagonizes the anti-apoptotic Mcl-1 and BCL-xL, reduces cell viability alone and in combination with retinoids in HL-60 cells [105]. In cells with non-functional p53 such as NB4 APL cells, Nutlin-1, the small-molecule inhibitor of MDM2, enhances ATRA-induced differentiation [106]. The authors demonstrated that the activity of Nutlin-1, combined with ATRA, depends on the competitive binding of Nutlin-1 to p-gp, leading to ATRA efflux inhibition and subsequent activation of differentiation pathways. Table 4 lists pre-clinical studies of Bcl-2 inhibitors and MDM2 inhibitors.

### 3.7. Combination of ATRA with Proteasome Inhibitors

ATRA induces degradation of RARs via the ubiquitin/proteasome pathway [114]. Two pre-clinical studies of proteasome inhibitors have been reported. Ying et al. [115] showed that bortezomib, a proteasome inhibitor developed for multiple myeloma, demonstrates strong synergistic effects with ATRA in promoting differentiation of HL-60 and NB4 cells. Fang et al. [116] reported that the differentiating effect of ATRA and proteasome inhibitors, which protect RARα from degradation, may be associated with the ubiquitin-proteasome pathway.

### 3.8. Combination of ATRA with Cytokines

Studies showed that granulocyte colony stimulating factor (G-CSF) potentiates ATRA-induced granulocytic differentiation in the APL cell line, various AML cell lines [107,108,109], and in MDS marrow [110]. One report revealed that the combination of G-CSF, ATRA, and the low dose cytotoxic drug cytarabine ocfosfate (a prodrug of araC) led to complete remission (CR) in a 67-year old patient with AML (M2) [112]. Another clinical success of G-CSF and ATRA was previously reported in a 61-year old man with APL [113]. The activation of the JAK-STAT [107] and MAP kinase [117] pathways is likely essential for inducing this differentiation. Another group demonstrated that ATRA combined with granulocyte macrophage colony stimulating factor (GM-CSF) induces differentiation in APL cells [108] and morphological changes in AML ML-1 cells [111]. The studies of ATRA combined with cytokines are limited, with two case reports and five pre-clinical studies, listed in Table 4. In addition, the limitations of cytokines is their potential to stimulate the growth of AML cells [118].

## 4. Novel Molecular and Natural Differentiation Enhancers

### 4.1. Combination of ATRA with Glycosylation Modifiers

Aberrant glycosylation, resulting from dysregulated glycan-related gene expression, is a hallmark of AML and affects cell adhesion, signaling, and disease subtype, serving as a potential diagnostic and prognostic biomarker. Targeting glycosylation pathways—including glycosyltransferase inhibitors, glycomimetics, and combination therapies—may enhance treatment efficacy [119]. We previously showed that the combination of the fucosylation inhibitor 6AF and ATRA enhances differentiation [120]. Modulation of glycosylation may represent a potential approach to enhance myeloid differentiation. However, pre-clinical and clinical studies investigating glycosylation modifiers remain limited (Table 5).

The natural (-)-Δ^9^-tetrahydrocannabinol isomer dronabinol is used to treat nausea and vomiting caused by chemotherapy. Kampa-Schittenhelm et al. [121] observed potent ex vivo cytoreductive sensitivity of T-lymphoblastic lymphoma cells to dronabinol. The O-linked β-N-acetylglucosamine transferase (OGT) is a master regulator enzyme that adds O-GlcNAc to serine or threonine residues in multiple target proteins [122]. Aberrant O-GlcNAc modification is implicated in the pathologies of metabolic and neurodegenerative diseases and cancers [123]. O-GlcNAcylation regulates transcription, cell signaling, cell division, cytoskeletal regulation, and metabolism [124]. One report showed that O-GlcNAc homeostasis contributes to cell fate decisions during early hematopoiesis [125]. Kampa-Schittenhelm et al. [121] found that dronabinol activates OGT, leading to differentiation of AML blasts in vitro and in vivo. Sassano et al. [126] demonstrated that statins exhibit antileukemic properties in vitro, suggesting potential synergy with ATRA. Fluvastatin inhibits FLT3 glycosylation [127]; thus, glycosylation may contribute to differentiation effects. Although reports on glycosylation modifiers are limited, they represent promising candidates for novel differentiation therapy because some are used clinically [121,126], with clinical effectiveness [121].

**Table 5 curroncol-33-00025-t005:** Studies of glycosylation modifiers. Early clinical refers to early-phase or exploratory clinical studies (Phase I–II, pilot, retrospective, or case report). Pre-clinical indicates studies limited to experimental models.

**Pre-Clinical Studies of Glycosylation Modifiers**					
**Differentiation Agent(s)**		**Action**		**Model Level**	**Ref**
6-alkynylfucose (6-AF) (fucosylation inhibitor)		ATRA or 6AF alone reduced fucosylation, whereas their combination produced a more pronounced decrease. Both 6AF and ATRA also synergistically enhanced differentiation in NB4 (APL) and HL-60 (AML) cells.		Cell line (NB4 and HL-60 cells).	[120]
Dronabinol (inducers of O-linked β-N-acetylglucosamine transferase)		Dronabinol, used to treat chemotherapy-induced nausea and vomiting, induced activation of O-linked β-N-acetylglucosamine transferase, resulting in differentiation of AML blasts in vitro and in vivo.		Cell line/primary blasts (Jurkat, MOLM14, primary AML cells).	[121]
Atorvastatin, Rosuvastatin, Fluvastatin. (Inhibitors of the 3-hydroxy-3-methylglutaryl-CoA reductase, which regulates not only cholesterol, but dolichol and ubiquinone. Dolichol mediates glycosylation.)		It was demonstrated that atorvastatin and fluvastatin effectively induced differentiation and apoptosis in NB4 APL cells, an effect regulated by Rac1/Cdc42 activation and its downstream c-Jun N-terminal kinase (JNK) signaling.		Cell line/primary blasts (NB4, RA resistant variants NB4.007/6, NB4.300/6, bone marrow or peripheral blood from patients with AML [AML-M2, AML-M5, and unclassified relapsed AML]).	[126]
**Clinical Study of Glycosylation Modifier**					
**Differentiation Agent(s)**	**Status**	**Patients Number**	**Dose and Schedule**	**Results**	**Ref**
Dronabinol	Early clinical (case report).	A 90-year old patient with AML	Hydroxyurea (HU, 2–3 × 1 g) was initially given for leukocytosis (~1.0 × 10^5^/µL, 80% blasts) and tapered as neutrophils declined. Dronabinol 2.5% was added, titrated to 6 drops twice daily.	Leukocytosis resolved, HU was discontinued, and dronabinol maintained. Peripheral blood blasts nearly disappeared, and neutrophil and platelet counts normalized.	[121]

### 4.2. Natural Products with Differentiation-Enhancing Effects

Numerous differentiation enhancers extracted from natural products have been reported (Table 6). All of these enhancers remain at the pre-clinical level. It is reported that, (-)-Epigallocatechin-3-gallate (EGCG), the major active polyphenol extracted from green tea, induces differentiation in leukemia cells. This is demonstrated using in an APL experimental mouse model [128] and in myeloid leukemia cell lines (NB4, HL-60) [129]. Another study elucidates the molecular mechanism by which EGCG enhances ATRA-induced differentiation in the APL cell line. It reveals a significant increase in the level of the tumor suppressor phosphatase and tensin homolog (PTEN) levels, when EGCG is combined with ATRA treatment in several leukemia cell lines. This synergy promotes the degradation of PML/RARα, restores PML expression, and increases the level of nuclear PTEN [130].

Dihydromyricetin (DMY), a 2,3-dihydroflavonol compound, exhibits a strong synergy with ATRA to promote differentiation in the APL NB4 cell line [131]. Wogonin, a flavonoid extracted from *Scutellaria baicalensis*, demonstrates significant anticancer activities [132] and potential therapeutic effects for hematologic malignancies [133]. Additionally, wogonoside, a metabolite of wogonin, induces cell cycle arrest and differentiation of AML cells [134,135]. Jiyuan oridonin A (JOA), a kaurene diterpenoid compound isolated from *Isodon rubescens,* inhibits cell proliferation and promotes differentiation, not only in AML cell lines [136], but also in CML cells with a *BCR::ABL* mutation [137].

Several other synthetic analogs, such as OGP46, are identified as myeloid differentiation inducers [138,139]. Recently, Parsa et al. [140] demonstrated that Silymarin, an anti-cancer herbal substance extracted from milk thistle (*Silybum marianum*), in combination with ATRA, enhances apoptosis and differentiation in human APL NB4 cells. Similarly, Gu et al. [141], identified a novel natural ent-kaurene diterpenoid derived from *Isodon pharicus* leaves, called pharicin B, which has a synergistic or additive differentiation-enhancing effect when used in combination with ATRA in several AML cell lines and primary APL cells. They further elucidate that pharicin B stabilizes RARα protein in the presence of ATRA, which is known to induce the degradation of RARα. Notopterol, a type of coumarin extracted from *Nardostachys jatamansi* (*N. incisum*), is known for its fever-relieving, analgesic, and anti-inflammatory effects [142]. Huang et al., [143] demonstrated that the combination of notopterol and ATRA enhances differentiation in APL cells, as evidenced by an increased nucleocytoplasmic ratio, NBT-positive cells, and the expression of CD14.

Fucoidan, a natural substance derived from marine algae, has been investigated for its potential anti-cancer effects [144]. Atahrazm et al. [145], reported that fucoidan enhances the therapeutic effects of ATO and ATRA in APL, but not in Kasumi-1 AML cells. The combination of fucoidan + ATRA or fucoidan + ATO delays tumor growth and induces differentiation in APL-bearing mice. Cotylenin A (CN-A), originally isolated as a plant growth regulator, shows potent differentiation-inducing effects in several AML cell lines and primary cells [146,147]. The compound affects several physiological processes in higher plants [148] and continues to exhibit strong differentiation inducing properties in AML cell line and primary cells [146,147,149,150]. Emodin, a natural compound extracted from the root of *Rheum palmatum*, exhibits multiple biological activities, including anticancer functions [151,152]. Chen et al. [153], demonstrated that emodin significantly enhances sensitivity to ATRA and has additive differentiation-inducing effects in the APL cell line NB4, especially in the NB4-derived ATRA-resistant MR2 cells. Hagiwara et al., examined the effect of ellagic acid (EA), a natural polyphenolic compound with antiproliferative and antioxidant properties, on HL-60 cell differentiation. They found that EA not only induces apoptosis but also potentiates ATRA-induced myeloid differentiation of HL-60 cells [154]. Gupta et al. [155], identified a natural compound securinine, a plant-derived alkaloid that has previously been used clinically as a therapeutic for primarily neurologically related diseases [156], which induces monocytic differentiation of a wide range of myeloid leukemia cell lines as well as primary leukemic patient samples. Jasmonic acid and its methyl ester, methyl jasmonate (MJ), which are fatty acid-derived cyclopentanones found in plants, are reported to exhibit anticancer activity in vitro and in vivo [157]. MJ induces differentiation in human myeloid leukemia cells [158,159]. Genistein, an isoflavone present in soybeans and other legumes, is known for its antitumor activity [160] and induces in vitro differentiation in several tumor cell models, including leukemia cells [161,162]. Asou et al. [163] showed that, resveratrol (3,4,40-trihydroxy-trans-stilbene), a phytoalexin found in grapes and other foods, inhibits proliferation and induces differentiation in various AML cell lines, including HL-60, NB4, U937, THP-1, ML-1, Kasumi-1 cells. Caffeic acid (CA), a widely distributed plant phenolic compound, enhances ATRA-induced differentiation, although CA alone has minimal or no effect on cell differentiation in HL-60 cells [164].

PC-SPES is a patented mixture of eight herbs, has been shown to be active against prostate cancer in vitro and in vivo [165]. Ikezoe et al. [166], demonstrated that PC-SPES decreases proliferation and induces differentiation and apoptosis in HL-60, NB4, U937 and THP-1 human myeloid leukemia cells. Furthermore, they show that PC-SPES enhances the antiproliferative and prodifferentiative effects of ATRA on these cells.

It was also demonstrated that plant redifferentiation-inducing hormones, cytokinins such as kinetin, isopentenyladenine, benzyladenine, and cotylenin A, potently induce granulocytic differentiation in myeloid leukemia HL-60 cells.

Finally, vibsane-type diterpenoids, specifically vibsanin A, derived from natural plant sources, are reported to induce differentiation of myeloid leukemia cells through PKC, and Raf/MEK/ERK activation. Vibsanin A induces monocytic differentiation in HL-60 cells, megakaryocytic differentiation in CML cells, and differentiation in 10 out of 11 primary AML patients, in a concentration-dependent manner [167].

**Table 6 curroncol-33-00025-t006:** Studies of natural products.

**Studies of Natural Products**				
**Differentiation Agent(s)**	**Primary Raw Material**	**Results**	**Model Level**	**Ref**
(-)-Epigallocatechin-3-gallate (EGCG)	Major active polyphenol extracted from green tea	In PML/RARα mice, EGCG administration reversed anemia, leukocytosis, and thrombocytopenia, and prolonged survival. In NB4 cells, EGCG upregulated neutrophil differentiation markers (CD11b, CD14, CD15, CD66) and, together with N-acetyl-L-cysteine (NAC), inhibited ROS production.	Cell line (APL model mice, NB4 cells).	[126]
		Treatment with EGCG significantly upregulated death-associated protein kinase 2 (DAPK2), accompanied by increased cell death in AML cells.	Cell line (HL60, NB4, retinoic-acid resistant NB4-R2 and HL60-R411).	[129]
		Treatment with EGCG and ATRA markedly upregulated PTEN in HL-60, NB4, and THP-1 cells, paralleled by increased CD11b expression. The combination synergistically facilitated PML/RARα degradation, restored PML expression, and elevated nuclear PTEN levels.	Cell line (HL-60, NB4 and THP-1).	[130]
Dihydromyricetin (DMY), a 2,3-dihydroflavonol compound	The main bioactive component extracted from Ampelopsis grossedentata	DMY sensitized NB4 cells to ATRA-induced growth inhibition, NBT reduction, CD11b expression, and upregulation of myeloid regulators (PU.1, C/EBPs). The DMY-enhanced differentiation appeared independent of PML-RARα and was mediated via activation of the p38–STAT1 signaling pathway.	Cell line (NB4 cells).	[131]
Wogonin (5,7-dihydroxy-8-methoxyflavone)	Monoflavonoid extracted from *Scutellariae radix*, a traditional Chinese medicine Huang-Qin	Wogonin promotes apoptosis in malignant T cells and inhibits growth of human T-cell leukemia xenografts. Importantly, normal T lymphocytes are largely unaffected, which is attributed to differential redox regulation in malignant versus normal cells.	Cell line (Malignant T-cell lines CEM, Molt-4, DND-41, JurkatJ16, J16neo, J16bcl-2, Jurkat A3, Jurkat A3 deficient in FADD, Jurkat cells deficient in LAT, SLP76 and PLCγ1).	[126]
		In U937 and HL-60 cells, wogonin suppressed proliferation via G1-phase arrest and induction of differentiation. Wogonoside significantly enhanced PLSCR1 transcription, accompanied by modulation of differentiation- and cell cycle-related genes, including increased p21 Waf1/Cip1 and decreased c-Myc expression.	Cell line/primary blasts (3 primary leukemic cells from AML patients, U937 and HL-60 cells).	[134]
		Wogonoside increased PLSCR1 expression and its binding to the 1, 4, 5-trisphosphate receptor 1 (IP3R1) promoter in primary AML cells. Activation of IP3R1 by wogonoside promoted Ca^2+^ release from the endoplasmic reticulum, contributing to cell differentiation.	Primary blasts/in vivo (23 Primary leukemic cells from newly diagnosed AML patients without prior therapy, U937 xenografts mice model and primary AML).	[135]
Jiyuan oridonin A (JOA), kaurene diterpenoid compound	Isolated from Isodon rubescensin	JOA markedly suppressed proliferation and induced differentiation, associated with G0/G1 cell-cycle arrest and impaired colony-forming ability.	Cell line (MOLM-13, MV4-11 and THP-1).	[136]
		JOA inhibits proliferation and induces G0/G1 cell-cycle arrest and differentiation in both imatinib-sensitive and -resistant CML cells, including those with the BCR-ABL-T315I mutation, by suppressing BCR-ABL/c-MYC signaling.	Cell line (Human K562 cells [BCR-ABL-native CML], murine BaF3 cells carrying wild-type p210 BCR-ABL [BaF3-WT] and point mutations of p210 BCR-ABL [T315I, E255K, G250E, M351T, Y253F, F359V, E255V, H296P, Q252H, F311L, M244V and F317L]).	[137]
Silymarin (SM)	Extracted from milk thistle (Silybum marianum)	Treatment with SM suppressed proliferation and potentiated ATRA-induced apoptosis in NB4 cells.	Cell line (NB4 cells).	[140]
Pharicin B	Natural entkaurene diterpenoid derived from Isodon pharicus leaves	Pharcin B induces myeloid differentiation in combination with ATRA in several AML cell lines and primary leukemia samples, enhancing ATRA-dependent transcriptional activity of RARα, which contributes to this effect.	Cell line/primary blasts (12 primary AML patients, U937, THP-1, NB4, and NB4-derived ATRA-resistant cell lines NB4-MR2, NB4-LR1, and NB4-LR2, as well as NB4FLAG-RARα and U937FLAG-RARα cell lines with stable expression of FLAG-RAR-α.	[141]
Notopterol	One type of coumarin, is an active monomer extracted from *N. incisum*	Inhibited the growth leukemia cells (IC50 [µM]: HL-60; 40.32 µM, Kasumi-1; 56.68, U937; 50.69) Notopterol also induced differentiation and G0/G1 arrest in HL-60 cells.	Cell line (HL-60, Kasumi-1, U937 cells).	[143]
Fucoidan	A natural substance derived from marine algae	Fucoidan induced apoptosis at 20 µg/mL in APL (NB4) but not in non-APL (Kasumi-1) cells when combined with ATO. In NB4 cells, fucoidan with ATO and/or ATRA efficiently promoted differentiation, and fucoidan plus ATRA or ATO delayed tumor growth while inducing differentiation.	Cell line/in vivo (NB4, Kasumi-1, APL-bearing mice).	[145]
Cotylenin A (CN-A)	Isolated from the metabolites of a simple eukaryote, a *cladosporium* sp. as plant growth regulators	CN-A efficiently induced differentiation in myeloid cell lines (HL-60, NB4, NB4/R, ML-1, and HT-93). In NB4 cells, CN-A promoted monocytic differentiation, as indicated by increased α-naphthyl acetate esterase activity.	Cell line (ML-1, HT-93, U937, TSU, P39/Fuji, JOSK-M, HL-60, NB4, retinoid-resistant NB4 [NB4/R]).	[126]
		CN-A induced differentiation in 9 of 12 primary patient samples. Synergistic effects were observed when CN-A was combined with ATRA (3 of 12) or vitamin D3 (8 of 12).	Primary blasts (12 primary AML specimens).	[148]
Emodin	Extracted from the root and rhizome of *Rheum palmatum* L.	Emodin sensitized ATRA induced differentiation in NB4, MR2 and primary AML samples. Emodin potently inhibits phosphorylation of Akt and efficiently inhibits mTOR downstream targets.	Cell line/primary blasts (NB4, MR2 [ATRA resistant NB4], 21 primary AML specimens).	[153]
Ellagic acid (EA)	EA is a polyphenolic compound found in fruits and berries	EA induced apoptosis and differentiation of HL-60 cells. In addition, EA sensitized ATRA induced differentiation.	Cell line (HL-60, NB4 cells).	[154]
Securinine	Major alkaloid natural product from the root of the plant *securinega suffruticosa*.	Securinine promoted monocytic differentiation in HL-60 and THP-1 cells and in several AML and one CML primary sample. It also significantly inhibited proliferation at 10–15 µM in various cell lines and in HL-60 xenograft models.	Cell line/primary blasts/in vivo (HL-60, THP-1, OCI-AML3, MV411, NB4, Nomo, U937 cells. 6 primary AML and CML samples. HL-60 xenografted mice).	[155]
Methyl jasmonate (MJ)	Jasmonates are potent lipid regulators in plants	MJ at 0.4 mM effectively upregulated CD15 and CD14, but not CD33, in HL-60 cells, inducing granulocytic differentiation with partial monocytic characteristics.	Cell line (HL-60, THP-1).	[158]
		Microarray analysis revealed that MJ, isopentenyladenine, and CN-A, but not vitamin D3 or ATRA, induced expression of the calcium-binding protein S100P. The MJ derivative, methyl 4,5-didehydrojasmonate, was 30-fold more potent than MJ.	Cell line/primary blasts (8 primary AML specimens, HL-60).	[159]
Genistein	Identified as the predominant isoflavone in soybean.	10 µg/mL of genistein efficiently reduced cell numbers, induced expression of OKM1 in HL-205 and benzidine positive cells in K-562-J, respectively.	Cell line (HL-205 [derivative of HL-60], K-562-J [derivative of K-562]).	[161]
		Genistein (10–25 µM) induced CD11b expression and G2/M cell-cycle arrest, effects that were enhanced by ATRA. MEK/ERK activation and accumulation of reactive oxygen species contributed to genistein-induced differentiation.	Cell line (HL-60, NB4 cells).	[162]
Resveratrol	A phytoalexin found in grapes and other food products.	Resveratrol (10 µM, 3 days) promoted CD11b expression in HL-60, NB4, U937, THP-1, and ML-1 cells, with additive effects observed upon co-treatment with ATRA or vitamin D3. At 20 µM, it induced NBT reduction and morphological differentiation in 8 of 19 primary leukemia samples.	Cell line/primary blasts (HL-60, NB4, U937, THP-1, ML-1, Kasumi-1, 19 primary AML, MDS and ALL samples).	[163]
Caffeic acid (CA)	A phenolic plant compound	ATRA-induced differentiation was potentiated by CA, with NBT reduction assays demonstrating an additive effect.	Cell line (HL-60 cells).	[164]
PC-SPES	Patented mixture of eight herbs	PC-SPES suppressed growth and promoted differentiation in HL-60 and NB4 leukemia cells, but enhanced proliferation of normal myeloid-committed CFU-GM cells.	Cell line (HL-60, NB4, U937 and THP-1 cells).	[166]
Vibsanin A	A vibsane-type diterpenoid isolated from the leaves of *Viburnum odoratissimum*	Vibsanin A promoted monocytic differentiation in HL-60 cells, megakaryocytic differentiation in CML cells, and induced differentiation in 10 of 11 primary AML samples in a concentration-dependent manner (maximum 10 µM). In mouse xenograft models, it extended host survival, effects mediated via protein kinase C activation and the Raf/MEK/ERK signaling pathway.	Cell line/primary blasts/in vivo (HL-60, U937 and NB4 cells, Xenografted mice [Injected cells were from spleen of leukemic Mll-AF9 transgenic mice, or HL-60 cells]. 11 primary AML cells).	[168]
**Inducers Those Can Serve As Antibiotics**				
**Differentiation Agent(s)**	**Primary Raw Material**	**Results**	**Model Level**	**Ref**
Nargenicin	Identified from the novel actinomycete strain CS682	Nargenicin (200 µM) induced differentiation in HL-60 cells and enhanced differentiation induced by vitamin D3 and ATRA. This effect was primarily mediated through the PKCβ1/MAPK pathways.	Cell line (HL-60 cells).	[126]
Deamino-hydroxy-phoslactomycin B (HPLM)	A biosynthetic precursor of phoslactomycin	HPLM induced differentiation in HL-60 cells via mechanisms distinct from those of ATRA and vitamin D3, which upregulate RARβ and 24OHase.	Cell line (HL-60 cells).	[169]
Salinomycin	A polyether ionophore antibiotic isolated from Streptomyces albus	Salinomycin combined with ATRA promoted differentiation by inhibiting β-catenin, which resulted in upregulation of PU.1 and C/EBPs and downregulation of c-Myc.	Cell line/primary blasts (Non-APL AML cells, primary AML cells).	[170]

### 4.3. Antibiotic-Derived Differentiation Enhancers

Several differentiation enhancers that also serve as antibiotics have been reported (Table 6). One such example is nargenicin, an unusual macrolide antibiotic previously identified from the novel actinomycete strain CS682, which exhibits strong antibacterial activity against methicillin-resistant *Staphylococcus aureus* [168]. Subsequently, the same group demonstrated that nargenicin induces differentiation in HL-60 cells when administered in combination with ATRA or 1,25-(OH)(2)D(3). Western blot analysis revealed that nargenicin enhances differentiation primarily through the PKCβ1/MAPK pathways [171]. Another compound, deamino-hydroxy-phoslactomycin B, a biosynthetic precursor of phoslactomycin, has been shown to induce myeloid differentiation in HL-60 cells [169]. Salinomycin, an antibacterial and coccidiostat ionophore therapeutic drug, selectively targets tumor cells over non-tumorigenic cells. One of its mechanisms of anti-tumor activity is the inhibition of the WNT/β-catenin pathway [170]. Salinomycin, when combined with ATRA, induces differentiation via WNT/β-catenin inhibition-mediated up-regulation of the myeloid transcription factor C/EBPs and PU.1, alongside suppression of c-Myc [172]. Furthermore, high levels of c-Myc expression in leukemic cells have been linked to resistance to chemotherapeutic drugs [173]. Consistent with this, Pan et al. [173], demonstrated that inhibition of c-Myc, either through shRNA or a specific c-Myc inhibitor 10058-F4, restores sensitivity to cytotoxic drugs, suppresses colony-forming ability, and promotes differentiation in leukemic cells.

### 4.4. Synthetic Small-Molecule Differentiation Inducers

Several myeloid differentiation inducers have been identified (Table 7), including the novel synthetic small compound LG-362B [174]. LG-362B induces differentiation in APL and ATRA-resistant APL cells as well as in an ATRA-sensitive/resistant transplantable mouse model. Treatment with 10mg/kg LG-362B significantly prolongs the survival of HL-60 xenografted mice. The mechanisms underlying the effects of LG-362B appear to depend on the degradation of PML-RARα regulated by caspase activity. 2-methyl-naphtho[2,3-b] furan-4,9-dione (FNQ3), a synthetic analogue of the quinone kigelinone, was initially identified as an agent effective for cancer cell death, with minimal effects on normal cells. It demonstrates cytotoxicity in various carcinoma cell lines, including prostate, cholangio, colon, laryngeal, and tongue cancers [175]. FNQ3 has also been reported to enhance the differentiation of HL-60 myeloid cells in the presence of either 1α, 25(OH)(2) dihydroxyvitamin D(3) [1alpha,25(OH)(2)D(3)] or ATRA [176]. NSC656243, a benzodithiophene compound, has been identified as an agent that amplifies ATRA-induced differentiation with acceptable cytotoxicity in NB4 cells. Even in the absence of ATRA, NSC656243 partially induces differentiation in HL-60 and murine erythroleukemia cells [177]. ST1346, the prototypical compound containing a bis-indolic structure (BISINDS), was identified by Pisano et al. and shown to enhance the differentiation effect of ATRA, both in cell lines and mouse models [178]. A new synthetic oleanane triterpenoid, 2-cyano-3,12-dioxoolean-1,9-dien-28-oic acid (CDDO), has been reported to exhibit potent differentiating, antiproliferative, and anti-inflammatory activities [179]. In particular, the C-28 methyl ester of CDDO, CDDO-Me, exhibits strong antiproliferative, apoptotic, and differentiating effects in leukemic cell lines and primary AML cells [180].

Several inhibitors of the DNA synthesis pathway have also been reported to induce differentiation. For example, some purine and pyrimidine analogs demonstrate differentiation-inducing effects. Honma et al., showed that novel synthetic neurotropic pyrimidine derivatives induce differentiation in human myeloid leukemia cells [181]. Additionally, a novel uracil analog, 6-chloro-5-(2-propenyl) uracil, has been reported to inhibit the growth and induce differentiation in myeloid leukemia cells [182]

Although high concentrations are required, several niacin-related compounds have also been reported to induce differentiation in HL-60 cells [183]. These include 6-aminonicotinamide (0.1 mM), 3-acetylpyridine (1 mM), nicotinic acid hydrazide (10 mM), nicotinamide (10 mM), and nicotinic acid (20 mM). While achieving effective concentrations in clinical applications may be challenging, the administration of niacin-related compounds shows no carcinogenic effect in mice [184]. Furthermore, prolonged survival has been observed in isonicotinic acid-treated mice [185].

**Table 7 curroncol-33-00025-t007:** Synthetic small molecules and repurposed agents enhancing ATRA-induced differentiation.

Synthesized Small Molecule Compound Inducers				
**Differentiation Agent(s)**	**Strategy for Identification**	**Results**	**Model Level**	**Ref**
LG-362B	A library of more than 100 synthesized compounds was screened for inhibition of APL cell proliferation.	LG-362B promoted differentiation in both APL and ATRA-resistant APL cells and in transplantable mouse models with ATRA-sensitive or resistant cells. Administration of 10 mg/kg LG-362B to HL-60 xenografted mice markedly extended survival, likely through caspase-dependent degradation of PML-RARα.	Cell line/in vivo (HL60, NB4, ATRA resistant NB4-R1 cells. HL-60 xenografted tumor mouse model, ATRA-sensitive/resistant transplantable mouse model).	[126]
2-Methyl-naphtho[2,3-*b*] furan-4,9-dione (FNQ3)	To obtain agents most efficient for cancer cell death with minimal effects for normal cells, FNQ3 was initially identified by Hirai et al. [175].	FNQ3 induced growth arrest and apoptosis in various human AML (HL-60, NB-4, U937, THP-1) and myeloma cell lines (RPMI-8226, ARH-77, NCI-H929, U266). Among primary AML samples, 11 of 14 showed reduced clonogenic growth.	Cell line/primary blasts (HL-60, NB-4, U937, THP1, RPMI-8226, ARH-77, NCI-H929, U266. 14 primary AML patients).	[176]
Benzodithiophenes (NSC656243)	Using ATRA insensitive NB4 cells (NB4-c) and NBT assay, 371 cytostatic agents from National Cancer Institute library were screened.	NSC656243 potentiated ATRA-induced differentiation in ATRA-insensitive NB4-c cells and induced dose- and time-dependent apoptosis in both NB4-c and HL-60 cells. Derivatives NSC656240, NSC656238, and NSC682994 further enhanced differentiation in NB4-c cells (NBT^+^%, dose in µM: NSC656243, 53, 5–7; NSC656240, 46, 0.05; NSC656238, 50, 0.05; NSC682994, 50, 0.01).	Cell line (NB4, NB4-c, HL-60, MEL cells, HL-60/Bcl-2 and HL-60/neo cells).	[177]
ST1346, ST1707 (a novel class of agents with bis-indolic structures (BISINDs)	Screening experiment using NB4 cells to select compounds that enhance the differentiating activity of ATRA	BISINDs augmented ATRA-induced STAT1 activation in APL cells and counteracted ATRA-mediated downregulation of Jun N-terminal kinases (JNK). This JNK activation likely contributes to the enhanced differentiation. Furthermore, ST1346 increased NBT-reducing activity across all examined cell lines.	Cell line (NB4, NB4.306, U937, Kazumi, HL-60, KG1, and PR9 [a U937-derived cell clone expressing PML-RARα upon induction with zinc sulfate] cells).	[178]
Oleanane triterpenoid 2-cyano-3,12-dioxoolean-1,9-dien-28-oic acid (CDDO)	Triterpenoids and some like ursolic and oleanolic acids are known to be anti-inflammatory and anticarcinogenic. The authors synthesized novel oleanane triterpenoid which has potent biological activities.	The compound promotes differentiation in diverse cell lines, including myeloid leukemia cells, and exhibits growth-inhibitory effects on a range of human tumor cell lines. It also downregulates pro-inflammatory cytokines, including IL-1, IFN-γ, and TNF-α, thereby reducing the expression of inducible nitric oxide synthase (iNOS) and cyclooxygenase-2 (COX-2).	Cell line (MCF-7, MDA-MB-231, 21-MT-1, 21-MT-2, 21-NT, 21-PT, THP-1, U937, HL-60, NB4, AML 193, KG-1, ML-1, NT2/D1, A2058, MDA-MB-468, SW626, AsPc-1, CAPAN-1*e*).	[179]
CDDO-Me, a novel C-28 methyl ester of CDDO.	As CDDO was shown to have potent antiproliferative and differentiating activity, the activity of C-28 methyl ester form of CDDO was Examined.	CDDO-Me induced apoptosis and promoted granulo-monocytic differentiation in HL-60 cells, while inducing monocytic differentiation in primary AML cells. The combination of ATRA with CDDO-Me or the RXR-specific ligand LG100268 further enhanced these effects.	Cell line/primary blasts (HL-60, KG-1, U937, Jurkat, NB4. HL-60–doxorubicin-resistant cells (HL-60-DOX). U937/Bcl-2 and its vector control, U937/pCEP. 4 primary AML and 2 primary CML-BC patients’ samples).	[180]
6-aminonicotinamide, 3-acetylpyridine, Nicotinic acid hydrazide, Nicotinamide, Nicotinic acid, etc.	Niacin related compound	Induce differentiation from morphology, and also loss of non-specific esterase activity.	Cell line (HL-60 cells).	[183]

## 5. Repurposed Agents with Unclear Mechanisms of Action

Lastly, this section categorizes several intriguing therapeutic drugs that function as differentiation enhancers (Table 8). Tamoxifen, a selective estrogen receptor modulator, has been a pivotal drug in the treatment of hormone receptor-positive breast cancer [186]. To identify agents useful for differentiation therapy in APL, Adachi et al. [187], screened various compounds, including inhibitors of signal transduction pathways and physiologically active molecules, and identified tamoxifen. Although the precise mechanisms remain unclear, they demonstrated that tamoxifen effectively enhances the differentiation-inducing and growth-inhibitory effects of ATRA in NB4 and HT93 APL cells, as well as primary APL cells. Manzotti et al., found that amantadine [188], an antiviral and anti-Parkinson agent, induces monocyte/macrophage-like differentiation in several myeloid leukemia cell lines when used in combination with suboptimal concentrations of ATRA or 1α, 25-hydroxycholecalciferol. Although the mechanisms behind this enhancement remain largely unknown, they suggested that the induction of the vitamin D receptor by amantadine might play a role in this differentiation process.

Metformin, a widely used drug for treating diabetes, has also been shown to induce differentiation in APL cells [189]. Specifically, the addition of metformin to NB4 APL cells results in activation of the MEK/MAPK pathway, which promotes differentiation of these cells. Lithium chloride (LiCl), a drug commonly used to treat bipolar disorder, has been reported to cause sustained leukocytosis due to an increase in granulocyte production [190]. Consistent with this observation, studies have demonstrated that combining ATRA with LiCl results in synergistic differentiation of WEHI-3B D+ myelomonocytic leukemia cells [191,192]. Lithium salts are known to inhibit glycogen synthase kinase-3 beta (GSK-3β), a protein that plays a crucial role in tumorigenesis in various cancers [193]. In phase I trial, Ueda et al. [194], investigated the use of lithium and tretinoin for treatment of relapsed and nine refractory non-APL leukemia patients. The study reported immunophenotypic changes associated with myeloid differentiation in five patients. Furthermore, four patients achieved disease stability, with no increase in circulating blasts for over four weeks. The authors concluded that while this combination is well tolerated, its clinical activity remains limited in the absence of additional antileukemic agents.

**Table 8 curroncol-33-00025-t008:** Repurposed agents with unclear mechanisms of action, enhancing ATRA-induced differentiation.

Repurposed Agents with Unclear Mechanisms of Action, Enhancing ATRA-Induced Differentiation				
**Differentiation Agent(s)**	**Characteristics**	**Results**	**Model Level**	**Ref**
Tamoxifen	Selective estrogen receptor modulator	LG-362B promoted differentiation in both APL Tamoxifen markedly potentiated the differentiation-inducing and growth-suppressive effects of ATRA in NB4 and HT93 APL cell lines, as well as in primary APL cells.	Cell line/primary blasts (HL-60, NB4 and HT93 APL cells. Normal mouse bone marrow cells. One primary APL cells).	[126]
Amantadine	An antiviral and anti-Parkinson agent	Amantadine induced monocyte–macrophage-like differentiation in several myeloid leukemia cell lines when combined with suboptimal concentrations of ATRA or 1α,25-dihydroxycholecalciferol.	Cell line (HL-60, U937, Kasumi-1 cells).	[188]
Metformin	Agent for treating diabetes	Metformin treatment in NB4 APL cells activated the MEK/MAPK pathway, promoting their differentiation.	Cell line (Kasumi-1, SKNO-1, HL-60, KG-1a and NB4 cells).	[189]
lithium chloride (LiCl)	Agent used for manic-depressive patients	Lithium led to the enlargement of the total neutrophil mass and neutrophil production.	Primary samples (12 lithium treated patients).	[190]
		LiCl (5 mM) enhanced ATRA (3 µM)-induced effects in 50% of the patients tested.	Primary blasts (Primary specimens from 13 AML, 6 APL patients).	[191]
		LiCl was more effective than G-CSF. Combinations of ATRA with LiCl resulted in the synergistic differentiation of WEHI-3B D+ myelomonocytic leukemia cells	Cell line (WEHI-3B myelomonocytic leukemia cells).	[192]
		LiCl treatment induced immunophenotypic changes indicative of myeloid differentiation in five patients, and four of them achieved disease stability with no rise in circulating blasts for over four weeks.	Primary blasts (Nine relapsed, refractory AML patients with median age 65 [39,40,41,42,43,44,45,46,47,48,49,53,54,55,56,57,58,59,60,61,62,63,64,65,66,67,68,69,70,71,72,73,74,75,76,77,78,79,80,93,94] years were enrolled).	[194]

## 6. Conclusions and Future Directions

This review highlights the current research on various differentiation enhancers. A schematic outline of the effects of combinations of ATRA and differentiation enhancers discussed in this review is shown in Figure 1. Among these enhancers, the combination of ATRA and ATO represents the standard therapy for APL [5,6,7], while ATRA combined with epigenetic modifiers (e.g., valproic acid, 5-azacitidine, tranylcypromine) shows promising results [8,9,10,11,12,13,14,15,17] in clinical studies not only for APL, but also for AML. Pre-clinical studies reveal the potential of de novo nucleotide biosynthetic pathway inhibitors and DNA-damaging agents in combination with ATRA, with numerous studies published from the 1990s to the present [78,79,80,81,82,83,84,85,86,87,88,89,90,91,92]. These findings may contribute to the development of clinical studies and the establishment of efficient therapies in the future.

In the 1990s, the combination of 1α,25-dihydroxyvitamin D3 and ATRA attracted attention for its ability to promote differentiation [195]. However, a significant limitation of the clinical application of vitamin D analogs is the supraphysiologic dose required, which can cause systemic hypercalcemia [196]. Addressing these issues remains crucial before vitamin D3 can be used in clinical settings. Glycosylation modifiers also show promise as combinational agents for leukemia therapy [1], given that alterations in glycosylation patterns are a unique characteristic of malignancies [197].

Many differentiation enhancers from extracted natural products, antibiotics, synthesized small molecule compounds, and repurposed clinical drugs have been identified, with their efficacy reported in pre-clinical studies. Additionally, several FDA-approved drugs, such as ethacrynic acid, khellin, oxcarbazepine, and alendronate, have been identified as potential differentiation enhancers through screening in zebrafish embryos [198].

The enhancers categorized as “repurposed agents with unclear mechanisms of action”, have attracted attention for their therapeutic potential in myeloid leukemia. Clinical studies on these agents for other malignancies show promise. For example, although limited activity has been reported for the combination of ATRA with tamoxifen for breast cancer [199,200], the combination of alitretinoin (9-cis-retinoic acid) and tamoxifen is well tolerated and shows certain antitumor activities in metastatic breast cancer [201]. Furthermore, for hepatocellular carcinoma, a combination of ATRA, tamoxifen and vitamin E has been reported to improve the clinical outcomes and increase the survival rates [202].

Metformin remains a topic of debate. While some recent studies suggest no benefit for disease-free survival or overall survival [203], earlier research reports that metformin reduces cancer incidence and improves outcomes in clinical settings [204]. This aligns with in vitro studies showing that metformin inhibits cancer cell proliferation [205]. Consequently, further investigation into the combination of metformin and ATRA in myeloid leukemia is warranted.

Lithium, primarily used to treat bipolar disorders, is also being explored as an anti-cancer agent [193]. Similarly to metformin, lithium has been reported to reduce cancer risk in patients with bipolar disorder [206]. In summary, the author believes that repurposed agents, such as tamoxifen, metformin, and lithium, may serve as promising differentiation enhancers in the future. A better understanding of the mechanisms underlying myeloid differentiation could lead to the expansion of efficient differentiation therapies for AML patients.

Combinations of ATRA with arsenic trioxide or selected epigenetic modulators are the most clinically advanced, showing high efficacy in APL and promising results in AML/MDS; combinations with CDK/kinase inhibitors, nucleotide synthesis inhibitors, DNA-damaging agents, proteasome inhibitors, cytokines, glycosylation modifiers, natural products, and repurposed clinical drugs remain largely pre-clinical or in early-phase studies; certain agents such as vitamin D analogs are limited by pharmacologic toxicity, while repurposed drugs like tamoxifen, metformin, and lithium are under early translational investigation. This addition provides a concise and integrated overview of the spectrum of ATRA-based combinations, from established clinical regimens to experimental pre-clinical enhancers, and their potential for future clinical translation. Notably, outside of APL, most available data are derived from small, non-randomized, or exploratory studies, underscoring the need for prospective and well-controlled clinical trials.

Despite remarkable success in APL, extending differentiation therapy to other AML subtypes remains a key challenge. Future studies should focus on translating promising preclinical findings—particularly those involving synthetic small molecules, nucleotide synthesis inhibitors, glycosylation modifiers, and repurposed clinical drugs—into early-phase clinical trials. Establishing reliable biomarkers to monitor differentiation and therapeutic response will be critical for clinical development.

From a clinical perspective, integrating ATRA-based combinations with current treatment backbones warrants careful evaluation. Outside of APL, however, most ATRA-based combination strategies remain investigational and should currently be evaluated primarily within the context of well-designed clinical trials. In contemporary AML practice, lower-intensity regimens for older or unfit patients are largely centered on hypomethylating agent (HMA) plus venetoclax combinations [207]. In lower-intensity regimens, ATRA-based enhancers could potentially be incorporated into HDAC inhibitor +venetoclax schedules, particularly for patients with monocytic or differentiation-prone phenotypes [3,4]. Similarly, in FLT3-mutated AML, combining ATRA with FLT3 inhibitors may augment differentiation and reduce leukemic stem cell persistence, as suggested by pre-clinical studies [44]. Identification of biomarkers—such as RARα signaling activity, MAPK pathway status, FLT3-ITD allele burden, or glycosylation signatures—may help stratify patients most likely to benefit from these strategies.

Furthermore, optimizing pharmacokinetics and minimizing systemic toxicity, as seen with vitamin D analogs, are necessary to achieve safe and sustained differentiation in patients. Collaborative efforts between basic and clinical researchers will accelerate the identification of effective ATRA-based combinations and promote the integration of differentiation therapy into standard AML management.

## Figures and Tables

**Figure 1 curroncol-33-00025-f001:**
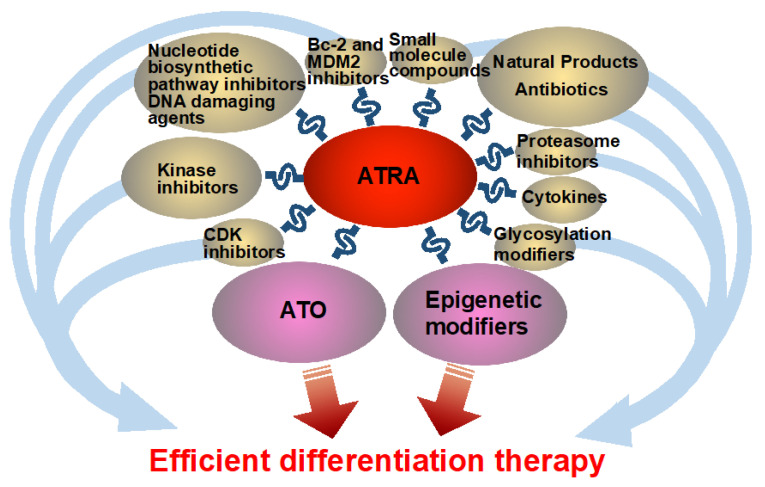
Schematic outline of the effect of combinations of ATRA and enhancers of differentiation described in this review. Pink circles indicate differentiation enhancers those effects (red arrows) are already approved at the clinical level, whereas yellow circles indicate enhancers those effects (light blue arrows) are shown mostly at the pre-clinical level. The size of the yellow circles reflects the number of studies. Curved connectors indicate combination strategies with ATRA.

**Table 1 curroncol-33-00025-t001:** Clinical ATRA-based Combination Therapies in AML and APL. Clinical denotes approved, guideline-based, or late-phase (Phase II–III) clinical studies. Early clinical refers to early-phase or exploratory clinical studies (Phase I–II, pilot, retrospective, or case report).

**Clinical Studies of ATO**					
**Differentiation Agent(s)**	**Status**	**Patients Number**	**Dose and Schedule**	**Results**	**Ref**
Oral arsenic	Clinical trial (Phase II)	54 high risk APL.	The consolidation treatment consisted of Realgar-Indigo naturalis formula (RIF) administered orally at 60 mg/kg per day in divided doses on a 4-weeks-on, 4-weeks-off schedule for a total of 4 cycles, along with ATRA given orally at 25 mg/m^2^ per day in divided doses on a 2-weeks-on, 2-weeks-off schedule for 7 cycles.	By the conclusion of the consolidation phase, all patients had attained molecular complete remission. Following consolidation therapy, two patients experienced relapse. Most adverse events observed were of grade 1 or 2 severity.	[5]
Oral arsenic, or ATO	Early clinical (Retrospective study)	212 patients with non-high-risk APL.	RIF was given orally at 60 mg/kg/day in three divided doses, or ATO was administered intravenously at 0.15 mg/kg/day, alongside ATRA at 25 mg/m^2^/day in two oral divided doses.	Five-year outcomes showed a relapse rate of 5.5%, with EFS and OS of 92.3% and 96.3%, respectively.	[6]
Oral arsenic, or intravenous ATO	Clinical (Phase III)	109 patients were enrolled and assigned to RIF-ATRA (n = 72) or arsenic trioxide-ATRA (n = 37).	Patients were randomly allocated (2:1) to receive induction and consolidation therapy with either RIF-ATRA or arsenic trioxide-ATRA. RIF (60 mg/kg/day orally in divided doses) or arsenic trioxide (0.15 mg/kg/day intravenously) was given in combination with ATRA (25 mg/m^2^/day orally in divided doses) until patients achieved complete remission.	At a median follow-up of 32 months, 2-year event-free survival was 97% in the RIF-ATRA group and 94% in the arsenic trioxide-ATRA group.	[7]
**Clinical Studies of Epigenetic Modifiers**					
**Differentiation Agent(s)**	**Status**	**Patients Number**	**Dose and Schedule**	**Results**	**Ref**
Valproic acid (VPA) followed by the addition of ATRA	Early clinical (Pilot study)	Eight high-risk or refractory AML patients not fit for intensive therapy	VPA treatment began on day 1 at 10 mg/kg/day orally, divided into three doses, with gradual titration to achieve serum concentrations of 50–110 µg/mL. ATRA (45 mg/m^2^/day orally in two divided doses) was added when therapeutic VPA levels were reached or by day 14 of treatment.	Using established MDS criteria, two patients showed hematologic improvement, one patient experienced disease progression, and five patients had stable disease.	[8]
VPA and ATRA	N/A	26 poor-risk AML patients	Oral ATRA (45 mg/m^2^) and VPA (5–10 mg/kg starting dose) were given, with hydroxyurea or low-dose AraC used to control leukocytosis.	No patients reached complete remission (CR). Two secondary AML patients from myeloproliferative disorders (MPD) achieved partial remission (PR) with peripheral blood blast clearance, and one de novo.	[9]
VPA, in combination with ATRA, or VPA alone	Clinical (Phase II)	75 patients with AML or MDS	VPA was titrated to maintain serum levels between 50 and 100 μg/mL, with a median treatment duration of 4 months. ATRA was administered at 80 mg/m^2^/day in two oral divided doses for a median of 2 months.	Response rates were 52% in myelodysplastic syndrome (MDS) with normal blast counts, 16% in AML, 6% in refractory anemia with excess blasts (I and II), and 0% in chronic myelomonocytic leukemia.	[10]
VPA	Early clinical (Pilot study)	11 elderly patients with de novo AML	VPA was given orally on a daily basis and adjusted to maintain therapeutic serum levels of 50–100 µg/mL. ATRA at 45 mg/m^2^/day was initiated one week after starting VPA. All patients received a minimum of one month of VPA–ATRA combination therapy.	Three patients demonstrated a complete marrow response; among them, one achieved CR and two showed hematologic improvement.	[11]
Low-dose AZA combined with pioglitazone, and ATRA	N/A	5 elderly AML patients	Treatment consisted of low-dose AZA (75 mg/day subcutaneously on days 1–7), PGZ (45 mg/day orally), and ATRA (45 mg/m^2^/day orally). Further AZA cycles were initiated upon hematologic recovery, with intervals of at least 28 days.	Of the 5 patients treated with APA therapy, 3 achieved ongoing morphological CR, and 2 of these also reached molecular CR.	[12]
Decitabine (DAC) alone, or DAC + VPA, or DAC + ATRA, or DAC + VPA + ATRA	Clinical (Phase II)	Two hundred patients unfit for induction chemotherapy.	The number of treatment cycles was 2 in the DAC-only arm (20 mg/m^2^ IV, days 1–5), 3 in the DAC + VPA arm, 5.5 in the DAC + ATRA arm, and 4 in the DAC + VPA + ATRA arm.	The addition of ATRA was associated with a higher remission rate (21.9% with ATRA vs. 13.5% without ATRA), whereas VPA showed no significant effect. Combining ATRA with DAC led to a clinically meaningful improvement in survival and remission rates.	[13]
AZA, VPA, and ATRA	Clinical (Phase II)	Sixty-five patients were enrolled.	Patients received oral VPA (35–50 mg/kg/day) and subcutaneous AZA (75 mg/m^2^/day) on days 1–7, followed by oral ATRA (45 mg/m^2^/day) on days 8–28. The treatment protocol consisted of six planned monthly cycles of AZA/VPA/ATRA.	The best responses comprised 14 CRs and 3 PRs (26%). Erythroid responses were observed in 75% of responders and 36% of non-responders. Median overall survival was 12.4 months, with untreated patients demonstrating longer survival than those with refractory or relapsed disease.	[14]
tranylcypromine (TCP)	Early clinical (Phase I/II)	18 patients with relapsed or refractory AML, unfit for intensive treatment.	Patients began TCP at 10 mg orally once daily on day 1, with daily increments of 10 mg as tolerated, up to a maximum of 60 mg/day over 7–10 days. ATRA (45 mg/m^2^/day orally) was started on day 7; however, if the TCP dose had not reached 50 mg/day, initiation of ATRA could be postponed until this threshold was achieved.	The overall response rate was 20%, comprising one partial response and two complete remissions without hematologic recovery.	[15]
TCP	Early clinical (Phase I)	17 patients with relapsed or refractory AML and MDS.	ATRA was given orally at 45 mg/m^2^ daily. TCP was administered at escalating dose levels of 10 mg, 20 mg, and 30 mg twice daily, beginning with a 3-day TCP monotherapy lead-in in cycle 1. Each treatment cycle was 21 days in the absence of dose-limiting toxicity and could be repeated if patients experienced clinical benefit.	The best responses included one hematologic improvement with complete remission of the marrow, one morphologic leukemia-free state, two cases of hematologic improvement with stable disease, and two cases of stable disease. The clinical benefit rate was 35.3%, and the overall response rate was 23.5%.	[17]
**Clinical Studies of Kinase Inhibitors**					
**Differentiation Agent(s)**	**Status**	**Patients Number**	**Dose and Schedule**	**Results**	**Ref**
Dasatinib (SFK inhibitor)	Early clinical (Phase I)	Nine subjects were enrolled in patients with high-risk myeloid neoplasms.	Three patients received ATRA at 45 mg/m^2^ combined with dasatinib 100 mg daily, while six patients received ATRA at 45 mg/m^2^ with dasatinib 70 mg daily, each for 28 days.	No clinical responses were observed.	[18]

**Table 2 curroncol-33-00025-t002:** Pre-clinical studies of ATRA with CDK inhibitors, kinase inhibitors and ATO. Pre-clinical indicates studies limited to experimental models.

**Pre-Clinical Studies of CDK Inhibitors**			
**Differentiation Agent(s)**	**Action**	**Model Level**	**Ref**
Palbociclib (CDK4/6 inhibitor)	AML cell proliferation was suppressed by palbociclib and ATRA.	Cell line/primary blasts (134 Primary AML blasts, HL-60, molm13)	[20]
Not applicable (palbociclib and ATRA could be effective)	Genome-scale transcriptome analysis in HL-60, NB4, and K562 cells identified CDK6 as a key regulator of the response to ATRA treatment.	Cell line (HL-60, NB4 and K562)	[21]
Roscovitine (CDK inhibitor)	Roscovitine together with ATRA induced nuclear phosphorylation of c-Raf, reinforced G1/G0 cell cycle arrest, and stimulated the expression of myeloid differentiation markers, including CD11b, reactive oxygen species, and p47 Phox.	Cell line (HL-60)	[23,24]
SU9516 (CDK2 inhibitor)	CDK2 inhibition activated differentiation and maturation pathways and significantly sensitized three AML subtypes to ATRA-induced cell differentiation.	Cell line/primary blasts (U937, NB4, HL-60, 3 primary AML blasts)	[25]
**Pre-Clinical Studies of Kinase Inhibitors**			
**Differentiation Agent(s)**	**Action**	**Model Level**	**Ref**
Sorafenib (multikinase inhibitor)	ATRA promotes p90RSK activation and GSK3β inactivation, which elevates Mcl-1 expression. Sorafenib counteracts this effect by preventing p90RSK activation and GSK3β inactivation, resulting in potentiation of ATRA-induced apoptosis.	Cell line (NB4 and its ATRA-resistance clone R4, HL-60, THP-1, ME-1, MOLM13, HL-60/Bcl2, Mcl-1 expressing HL-60/M15 cells).	[26]
Dasatinib (SFK inhibitor)	Dasatinib promotes ATRA-induced differentiation in AML cell lines and primary blasts via Lyn inhibition-mediated activation of the RAF-1/MEK/ERK pathway.	Cell line (ATRA resistant cell lines NB4-R1 and NB4-R2 cells) [27]. Primary blasts (31 Primary AML blasts) [28]. Cell line (HL-60 and NB4 cells) [29,30].	[28,29,30,31]
PP2 (SFK inhibitor)	The combination of PP2 with ATRA and/or ATO significantly upregulated ICAM-1 expression.	Cell line (NB4 cells)	[32]
Radotinib (BCR/ABL tyrosine kinase inhibitor)	Radotinib induced CD11b expression and promoted differentiation by suppressing Lyn. Apoptosis in CD11b^+^ cells was mediated by caspase-3 activation and mitochondrial membrane depolarization.	Cell line/primary blasts (HL60, THP-1, Kasumi-1, NB4, and primary AML cells).	[33]
Dasatinib (SFK inhibitor)	Dasatinib promotes differentiation by inducing autophagy.	Cell line/in vivo (NB4, U937, HL60, HL60 cells xenografted nude mice).	[34]
Enzastaurin (derivative of PKC pan-inhibitor staurosporine)	Enzastaurin combined with ATRA efficiently induced apoptosis through a mitochondria-dependent, but caspase-independent, pathway.	Cell line (ATRA-resistant APL cell lines, NB4-R1 and NB4-R2).	[35]
	The combination of enzastaurin and ATRA activated the MEK/ERK and Akt pathways while inhibiting PKCβ, leading to the upregulation of the myeloid transcription factors C/EBPβ and/or PU.1.	Cell line/primary blasts (HL-60, ATRA-resistant cell line, HL-60Res, and U937 as well as non-APL AML primary cells).	[36]
Staurosporine (PKC pan-inhibitor)	Activation of MEK/ERK by staurosporine promoted the expression of C/EBPβ and C/EBPε, thereby augmenting ATRA-induced differentiation, likely independent of PKC signaling.	Cell line (U937, K562 and Kasumi cells).	[37]
HG-9-91-01, YKL-05-099 (SIK inhibitors)	SIK inhibition augments ATRA-induced differentiation through a mechanism dependent on Akt pathway activation.	Cell line (HL-60, NB4, U937, and THP-1 cells).	[38]
LY294002 (PI3K/Akt inhibitor)	Ribosome profiling combined with transcriptome sequencing revealed that ATRA translationally regulates genes enriched in the PI3K/AKT signaling pathway. Inhibition of PI3K/AKT strongly induced apoptosis in AML cells.	Cell line/primary blasts/in vivo (Molm13, NB4, MV4-11, HL60, THP-1 cells. Molm13, NB4 engrafted mice. CD34+ HSPCs from the cord blood of healthy donors).	[39]
RAD001 (mTOR inhibitor everolimus)	ATRA (0.1–1 µM) combined with RAD001 (10 nM) significantly promoted differentiation through inhibition of mTORC1, leading to induction of C/EBPε and p27^Kip1, and downregulation of c-Myc.	Cell line (NB4, HL60 cells).	[40]
ZD1839 (EGFR inhibitor gefitinib)	Gefitinib, when combined with ATRA and ATO, promoted myeloid differentiation even in ATRA- and ATO-resistant APL cells, most likely through an off-target mechanism.	Cell line/in vivo (NB4 [ATRA-sensitive] and NB4-R2 [ATRA-resistant cells], APL transgenic mouse model).	[41]
Erlotinib and gefitinib (EGFR inhibitors)	Erlotinib suppressed the (auto)phosphorylation of p38 MAPK and SFKs, thereby mimicking the differentiation-inducing effects of EGFR inhibitors and enhancing ATRA- or vitamin D-induced differentiation.	Cell line/primary blasts (HL-60 and MOLM-13 cells, primary leukemic cells from 24 AML patients).	[42]
Trametinib (MEK inhibitor)	Co-treatment with trametinib and ATRA promoted the phosphorylation of Akt and JNK and upregulated STAT3 expression, leading to augmented differentiation.	Cell line/primary blasts (HL-60, U937, HL-60Res [ATRA resistant AML cells], and primary AML cells).	[43]
Sorafenib, AC220, TTT-3002 (FLT3 inhibitors)	In FLT3-ITD cells, FLT3 inhibitors synergize with ATRA to promote apoptosis. Although sorafenib induces Bcl6, which suppresses kinase inhibitor–mediated apoptosis, ATRA counteracts this suppression via STAT3-dependent signaling.	Cell line/primary blasts/in vivo (Molm14, MV4-11, THP-1, NB4, 7 primary AML samples, mouse xenograft models).	[44]
INCB52793 (JAK1 inhibitor)	Co-treatment with ATRA and a JAK1 inhibitor synergistically enhanced differentiation and growth suppression, accompanied by transcriptional changes indicative of differentiation, including the downregulation of G2/M checkpoint, E2F target, and MYC target gene sets.	Cell line/in vivo (MV-4–11, KG-1, K-562, HL-60, MOLM-13, Kasumi-1 and THP-1. MV-4-11 cell line-derived xenografts).	[45]
TAK165 (HER2 inhibitor)	RARα and STAT1 activation, triggered via phosphorylation of the MEK/ERK pathway, is essential for ATRA and TAK165–induced differentiation.	Cell line/primary blasts (HL60, HL60R, NB4, human breast cancer BT474 cells, 5 primary cells from AML patients).	[46]
**Pre-Clinical Studies of ATO**			
**Differentiation Agent(s)**	**Action**	**Model Level**	**Ref**
ATO	ATO plus ATRA robustly induced differentiation, markedly downregulating proteinase 3 and modulating azurocidin, telomerase reverse transcriptase, ferritin, and interleukin-1β (IL1B) compared with ATRA alone.	Cell line (HL-60)	[47]
2-D08, anacardic acid (pharmacologic inhibitors of SUMOylation)	In non-APL AML cell lines and primary cells, SUMOylation inhibitors upregulated ATRA-responsive genes (RARA, CEBPA, ITGAM, IL1B, etc.) associated with proliferation, apoptosis, and differentiation, resulting in both differentiation and cell death.	Cell line/primary blasts/in vivo (U937, HL60, THP1, MOLM14, U937-AraC resistant cells, U937 xenografted mice, 16 primary AML cells).	[48]
ATO	While ATO alone does not trigger differentiation in PML-RARα–negative HL-60 cells, it enhances ATRA-induced differentiation. ATO also amplifies ATRA-mediated activation of the RAF/MEK/ERK pathway.	Cell line (HL-60)	[49]

**Table 4 curroncol-33-00025-t004:** Pre-clinical studies of ATRA with apoptosis- and cytokine-related agents. Pre-clinical indicates studies limited to experimental models.

**Pre-Clinical Studies of Bcl-2 Inhibitors or MDM2 Inhibitors**					
**Differentiation Agent(s)**		**Action**		**Model Level**	**Ref**
ABT-737 (Bcl-2 inhibitor)		The efficacy of ABT-737 varied among seven AML cell lines, with IC50 values ranging from 9.9 nM to 1.8 µM. Sensitivity, evaluated via 18F-FDG uptake, correlated with ABT-737 activity. Furthermore, ABT-737 induced ATRA-mediated differentiation in NB4 and AML-193 cells.		Cell line (KG-1a, Kasumi-1, NB-4, PLB-985, MV4-11, THP-1 and AML-193).	[104]
JY-1-106 (Bcl-2 inhibitor), in combination with retinoids including ATRA, AM580 (RARα agonist), and SR11253 (RARγ antagonist)		The combination of the Bcl-2 inhibitor JY-1-106 and the RARγ antagonist SR11253 most effectively reduced cell viability by inducing apoptosis.		Cell line (HL-60 cells).	[105]
Nutlin-1 (MDM2 inhibitor)		Nutlin-1, an MDM2 inhibitor, significantly augmented ATRA-mediated differentiation in NB4 and HL-60 cells, which lack functional p53, but not in p53 wild-type U937 cells. Mechanistically, Nutlin-1 appears to competitively inhibit P-gp, reducing ATRA efflux and promoting activation of differentiation pathways.		Cell line (HL60, NB4, U937 and p53-silenced U937 cells).	[106]
**Pre-Clinical Studies of Cytokines**					
**Differentiation Agent(s)**		**Action**		**Model Level**	**Ref**
G-CSF		ATRA (100 nM) induced myeloid differentiation in HT93A cells, and this effect was enhanced by G-CSF, whereas G-CSF alone had no effect. In NB4 and THP-1 cells, G-CSF had minimal impact on ATRA-induced differentiation. In HT93A cells, G-CSF activated the JAK pathway, leading to STAT5 activation.		Cell line (HT93A [APL], NB4 [APL] and THP-1 [acute monocytic leukemia] cells).	[107]
G-CSF or GM-CSF		G-CSF or GM-CSF enhanced ATRA-induced differentiation in terms of morphology, reduced nitroblue tetrazolium (NBT) reduction, and increased expression of differentiation markers (CD11a, CD11b). Notably, only the combination of ATRA and G-CSF, but not GM-CSF, increased chemotactic activity.		Primary blasts (12 primary APL cells).	[108]
G-CSF		In the presence of ATRA, G-CSF further augmented differentiation through increased respiratory burst activity, while G-CSF alone was ineffective. G-CSF promoted terminal maturation of human bone marrow myeloid cells, concomitant with CD10 induction and G-CSF receptor-mediated signaling.		Cell line/primary blasts (NB4 cells, normal human bone marrow mononuclear cells).	[109]
G-CSF		In semi-solid cultures of normal bone marrow, G-CSF enhanced the size of granulocyte-macrophage (GM) colonies without affecting their number, whereas ATRA increased GM colony-forming cell (CFC) numbers.		Primary blasts (10 primary MDS bone marrow cells, and one normal bone marrow cells).	[110]
GM-CSF		Even at a high concentration (10^−7^ M), ATRA alone induced only a modest increase in NBT-reducing activity. However, in combination with GM-CSF, NBT-reducing activity was markedly enhanced. A synergistic effect of the two agents was also observed on inhibition of cell proliferation and morphological differentiation.		Cell line (ML-1 cells).	[111]
**Clinical Studies of Cytokines**					
**Differentiation Agent(s)**	**Status**	**Patients Number**	**Dose and Schedule**	**Results**	**Ref**
G-CSF, low-dose cytotoxic drugs	Early clinical (case report).	A 67-year-old patient with AML (M2)	ATRA (45 mg/m^2^, 70 mg daily) and cytarabine ocfosfate (SPAC, 50 mg daily) were administered alongside heparin for 36 days. Subsequently, SPAC was escalated to 200 mg daily in combination with ATRA and G-CSF for 18 days. Upon an increase in marrow blasts to 10%, oral cyclophosphamide (CPA, 50 mg daily) was added.	After 14 days of CPA combined with ATRA, G-CSF, and SPAC, marrow blasts declined to 1%.	[112]
G-CSF	Early clinical (case report).	A 61-year-old male with APL	Patients received ATRA (60 mg daily) for 14 days. Due to insufficient differentiation, G-CSF (75 µg/day) was added for 5 days.	CR was subsequently achieved, and treatment continued with daily ATRA and G-CSF twice weekly for a further 24 days.	[113]

## Data Availability

This review article did not generate any new data. All data supporting the findings discussed are available in the cited literature.

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
