# Peer review of "Reawakening Differentiation Therapy in Acute Myeloid Leukemia: A Comprehensive Review of ATRA-Based Combination Strategies"

_curroncol, 2026, doi:10.3390/curroncol33010025_

Round 1

Reviewer 1 Report

Comments and Suggestions for Authors

This review covers ATRA-based combination strategies in APL and non-APL AML and brings together a large amount of clinical and pre-clinical data. The topic is relevant, and the manuscript is generally well organized and clearly written. I have a few suggestions that I think would strengthen it further.

1. You mention a PubMed search yielding >500 papers up to November 2025. It would help readers if you briefly describe how you selected the studies you finally discussed (databases, time frame, basic inclusion criteria, types of studies). A short “Search Strategy and Study Selection” paragraph is sufficient – this does not need to be a full systematic review.

2. Since there are previous reviews on differentiation therapy (including your own earlier reviews and the Abdel-Aziz paper you cite), it would be useful to state explicitly in the Introduction what is new here – for example, broader coverage of combination partners, emphasis on mechanisms converging on ATRA/RARα, and more detailed clinical tables.

3. The clinical sections on ATRA+ATO and epigenetic modifiers are informative, but the tone is mostly descriptive. I suggest adding a short paragraph that clearly summarizes the main limitations: small and heterogeneous cohorts, modest responses in many non-APL AML studies, and lack of randomized or controlled trials for most combinations. This will give a more balanced view of how close these regimens are to routine practice.

4. You discuss established APL regimens, early clinical trials, and purely pre-clinical combinations in a similar way. To guide readers, please make the development stage more explicit in the text and especially in the tables (e.g., add a column such as “Status: approved / phase I–II / pre-clinical”). A short summary paragraph ranking which combinations are closest to clinical translation would also be helpful.

5. You nicely describe many pathways (MAPK/ERK, PI3K/Akt, JAK/STAT, SUMOylation, epigenetic changes, etc.) but most of this remains in text. I strongly encourage adding one schematic figure centered on ATRA/RARα signaling, showing where the main classes of combination partners act. This would help readers quickly see how different agents converge on common differentiation pathways.

5. The Conclusion could link more explicitly to current treatment backbones. For example, you might briefly discuss how ATRA-based combinations could be integrated with HMA+venetoclax regimens or with FLT3 inhibitor–based therapy, and whether there are potential biomarkers to select patients for these strategies.

Minor points 

1. The English is generally good, but there are a few minor issues with subject–verb agreement and spacing (e.g., “ATRA -based” → “ATRA-based”). A light language edit would fix these.

2. Ensure all abbreviations are defined at first use (e.g., RIF, APA, PGZ).

3. In the tables, consider standardizing response categories (CR, PR, HI, molecular response) and units; a separate column indicating the model type (cell line, primary blasts, in vivo) in pre-clinical tables would also improve clarity.

Author Response

Reviewer 1:

This review covers ATRA-based combination strategies in APL and non-APL AML and brings together a large amount of clinical and pre-clinical data. The topic is relevant, and the manuscript is generally well organized and clearly written. I have a few suggestions that I think would strengthen it further.

1. You mention a PubMed search yielding >500 papers up to November 2025. It would help readers if you briefly describe how you selected the studies you finally discussed (databases, time frame, basic inclusion criteria, types of studies). A short “Search Strategy and Study Selection” paragraph is sufficient – this does not need to be a full systematic review.

Thank you for this valuable suggestion. In response, I have added a concise paragraph as the penultimate paragraph of the Introduction section. This paragraph briefly outlines the literature search process, including the database used (PubMed), the time frame (inception–November 2025), search terms, inclusion criteria (ATRA-based combination studies relevant to myeloid differentiation, including clinical trials, in vivo studies, and mechanistic in vitro studies), and exclusion criteria (non-myeloid cancers, non-ATRA retinoids unless contextually relevant, and non-combination studies). This addition clarifies how the studies included in this review were selected (page 2, line 86).

A PubMed search from inception to November 2025 using the terms “ATRA,” “myeloid,” and “differentiation inducer or enhancer” identified over 500 articles. Studies were included if they examined ATRA-based combination strategies that promote myeloid differentiation in clinical trials, in vivo models, or mechanistic in vitro systems. Studies on non-myeloid cancers, non-ATRA retinoids, or single-agent ATRA without combination partners were excluded. Articles meeting these criteria were incorporated into this narrative review.

2. Since there are previous reviews on differentiation therapy (including your own earlier reviews and the Abdel-Aziz paper you cite), it would be useful to state explicitly in the Introduction what is new here – for example, broader coverage of combination partners, emphasis on mechanisms converging on ATRA/RARα, and more detailed clinical tables.

I agree and have revised the last paragraph of the Introduction to explicitly describe the novelty of this review. I added following sentences (page 3, line 93):

The principal novelty of this review is its comprehensive and unusually broad coverage of ATRA-based combination strategies. In addition to well-established partners such as arsenic trioxide and epigenetic modulators, this manuscript systematically integrates data on a wide spectrum of agents—including CDK and kinase inhibitors, nucleotide synthesis inhibitors, DNA-damaging agents, Bcl-2/MDM2 inhibitors, proteasome inhibitors, cytokines, glycosylation modifiers, natural products, and antibiotic-derived compounds. By compiling clinical findings together with pre-clinical in vivo and mechanistic in vitro studies across this diverse range of therapeutic classes, this review provides one of the most extensive and unified summaries to date of ATRA-enhanced myeloid differentiation.

3. The clinical sections on ATRA+ATO and epigenetic modifiers are informative, but the tone is mostly descriptive. I suggest adding a short paragraph that clearly summarizes the main limitations: small and heterogeneous cohorts, modest responses in many non-APL AML studies, and lack of randomized or controlled trials for most combinations. This will give a more balanced view of how close these regimens are to routine practice.

I thank the reviewer for this important suggestion. In response, I have added a paragraph at the end of the clinical sections (2.2) summarizing the main limitations of the current evidence on ATRA-based combinations for non-APL AML patients (page 7, line 167):

Clinical evidence is still preliminary, with small and heterogeneous cohorts, modest responses in non-APL AML, and few randomized trials. Larger studies are needed to confirm clinical benefit.

This addition provides a more balanced perspective on the current clinical status of ATRA-based combination therapies and highlights the need for larger, well-controlled trials to clarify their therapeutic potential.

4. You discuss established APL regimens, early clinical trials, and purely pre-clinical combinations in a similar way. To guide readers, please make the development stage more explicit in the text and especially in the tables (e.g., add a column such as “Status: approved / phase I–II / pre-clinical”). A short summary paragraph ranking which combinations are closest to clinical translation would also be helpful.

I thank the reviewer for this suggestion. In our tables summarizing clinical and pre-clinical studies, I have already included columns indicating the type of study (e.g., clinical, in vivo, in vitro) and the phase of trial (e.g., pilot study, phase II), which clearly denote the development stage of each combination.

Furthermore, I have added a short summary paragraph in the Discussion section highlighting the relative maturity of different strategies (page 33, line 722):

Combinations of ATRA with arsenic trioxide or selected epigenetic modulators are the most clinically advanced, showing high efficacy in APL and promising results in AML/MDS; combinations with CDK/kinase inhibitors, nucleotide synthesis inhibitors, DNA-damaging agents, proteasome inhibitors, cytokines, glycosylation modifiers, natural products, and repurposed clinical drugs remain largely pre-clinical or in early-phase studies; certain agents such as vitamin D analogs are limited by pharmacologic toxicity, while repurposed drugs like tamoxifen, metformin, and lithium are under early translational investigation. This addition provides a concise and integrated overview of the spectrum of ATRA-based combinations, from established clinical regimens to experimental pre-clinical enhancers, and their potential for future clinical translation.

5. You nicely describe many pathways (MAPK/ERK, PI3K/Akt, JAK/STAT, SUMOylation, epigenetic changes, etc.) but most of this remains in text. I strongly encourage adding one schematic figure centered on ATRA/RARα signaling, showing where the main classes of combination partners act. This would help readers quickly see how different agents converge on common differentiation pathways.

Thank you very much for this constructive suggestion. I fully agree that schematic pathway figures are valuable for illustrating mechanistic relationships among ATRA, RARα, and cooperating signaling pathways such as MAPK/ERK, PI3K/Akt, JAK/STAT and other regulators.

However, the principal novelty and purpose of the present review differ from my prior mechanistic reviews. As clarified in the revised Introduction, the main novelty of this manuscript is its unusually broad and comprehensive integration of ATRA-based combination strategies across a wide therapeutic spectrum, including not only well-established partners such as arsenic trioxide and epigenetic modulators, but also CDK/kinase inhibitors, nucleotide synthesis inhibitors, DNA-damaging agents, Bcl-2/MDM2 inhibitors, proteasome inhibitors, cytokines, glycosylation modifiers, natural products, and antibiotic-derived compounds.

Detailed mechanistic pathway illustrations centered on ATRA/RARα signaling have already been extensively presented in my previous reviews, including:

1.           Takahashi S. Current Understandings of Myeloid Differentiation Inducers in Leukemia Therapy. Acta Haematol. 2021;144:380-8.

2.           Takahashi S. Kinase Inhibitors and Interferons as Other Myeloid Differentiation Inducers in Leukemia Therapy. Acta Haematol. 2022;145:113-21.

Because detailed mechanistic figures are already presented in my previous reviews, I refer readers to those publications for pathway illustrations as follows (page 10, line 206).

Detailed mechanistic schematics of these pathways have been presented in my previous reviews [1, 2].

In this manuscript, I instead focus on integrating clinical, in vivo, and mechanistic in vitro data across the broadest spectrum of ATRA-enhancing agents. Adding another pathway figure here would largely duplicate existing material and detract from the manuscript’s primary purpose—providing a unified and clinically oriented synthesis of diverse ATRA-based combination strategies. To maintain clarity and avoid redundancy, I have therefore not included a new figure. The text now explicitly guides readers to my prior reviews for comprehensive mechanistic schematics. I hope the reviewer will agree that this approach preserves the coherence and intended scope of the manuscript while still ensuring access to the relevant mechanistic diagrams.

5. The Conclusion could link more explicitly to current treatment backbones. For example, you might briefly discuss how ATRA-based combinations could be integrated with HMA+venetoclax regimens or with FLT3 inhibitor–based therapy, and whether there are potential biomarkers to select patients for these strategies.

 Thank you for this constructive suggestion. In the revised Conclusions and future directions, I now briefly address how ATRA-based combinations may be positioned relative to current therapeutic backbones. Specifically, I note that HMA–venetoclax and FLT3 inhibitor–based regimens represent the contemporary standards in AML, and that ATRA-based differentiation strategies could ultimately complement—rather than replace—these backbones as biomarkers for differentiation response become better defined. This addition clarifies the potential clinical integration of ATRA-based combinations while maintaining the review's primary focus on differentiation-enhancing agents. The text has been updated accordingly (page 34, line 738).

From a clinical perspective, integrating ATRA-based combinations with current treatment backbones warrants careful evaluation. In lower-intensity regimens, ATRA-based enhancers could potentially be incorporated into HDAC inhibitor +venetoclax schedules, particularly for patients with monocytic or differentiation-prone phenotypes [3, 4]. Similarly, in FLT3-mutated AML, combining ATRA with FLT3 inhibitors may augment differentiation and reduce leukemic stem cell persistence, as suggested by pre-clinical studies [49]. Identification of biomarkers—such as RARα signaling activity, MAPK pathway status, FLT3-ITD allele burden, or glycosylation signatures—may help stratify patients most likely to benefit from these strategies.

Minor points 

1. The English is generally good, but there are a few minor issues with subject–verb agreement and spacing (e.g., “ATRA -based” → “ATRA-based”). A light language edit would fix these.

I thank the reviewer for the helpful suggestions. I have made minimal language corrections, including spacing (e.g., “ATRA-based”) and minor grammatical adjustments, while keeping the original meaning unchanged.

2. Ensure all abbreviations are defined at first use (e.g., RIF, APA, PGZ).

I appreciate the reviewer’s note regarding undefined abbreviations. I have carefully reviewed the manuscript and ensured that all abbreviations, including RIF, APA, and PGZ, are now defined at first use.

3. In the tables, consider standardizing response categories (CR, PR, HI, molecular response) and units; a separate column indicating the model type (cell line, primary blasts, in vivo) in pre-clinical tables would also improve clarity.

I thank the reviewer for this helpful suggestion regarding the standardization of clinical response categories and the inclusion of development-stage information in the tables. I fully agree that clarity in summarizing clinical and pre-clinical data is essential. In the current version of the manuscript, the clinical tables already list standard response categories (e.g., CR, PR) whenever these are explicitly reported in the original studies. For reports in which such standardized terms are not provided, the outcomes are described exactly as presented by the authors, to maintain accuracy and avoid over-interpretation.

For the pre-clinical tables, the information is organized into separate columns—Differentiation agent(s)Action, and Cells examined—which specify the model type (cell line, primary blasts, or in vivo) and clearly indicate the experimental context. Together with the inclusion of study type and trial phase in the clinical tables, these elements already distinguish the developmental stage of each combination.

Given these features, I believe the current table structure effectively conveys the intended information; nevertheless, I have rechecked the tables and made minor adjustments to ensure consistency and clarity.

Reviewer 2 Report

Comments and Suggestions for Authors

The article is stunning, very specific. Summarizes all the latest papers in the field of APL treatment and potential mechanisms of AML differentiatiom It is very informative. it would be easier if first use of abbreviation was accompanied with its explanation. Also I suggest adding a short information about APL and AML biology in the beginning

Author Response

Reviewer 2:

The article is stunning, very specific. Summarizes all the latest papers in the field of APL treatment and potential mechanisms of AML differentiatiom It is very informative. it would be easier if first use of abbreviation was accompanied with its explanation. Also I suggest adding a short information about APL and AML biology in the beginning.

I sincerely appreciate the reviewer’s positive evaluation of my manuscript. I agree that defining abbreviations at first use improves readability, and I have now ensured that all abbreviations are clearly explained upon initial appearance. I also appreciate the suggestion to provide brief background information on APL and AML biology. I have added a short introductory paragraph to give readers essential context before entering the main review (page 2, line 57).

APL is characterized by the promyelocytic leukemia (PML) – retinoic acid receptor alpha (RARα) fusion gene, which causes a maturation arrest at the promyelocyte stage, while AML more broadly is a heterogeneous group of myeloid malignancies defined by differentiation blocks at various developmental stages.

Reviewer 3 Report

Comments and Suggestions for Authors

Thank you for the opportunity to review this interesting paper. The manuscript presented a comprehensive review of ATRA in APL and beyond. Indeed, outside the APL setting, the authors clearly reported synergistic differentiation effects when ATRA is combined with several other compounds, including BCL-2 inhibitors and epigenetic modulators, with the potential for these therapeutic associations to be tested in clinical trials for the treatment of non-APL AML. The manuscript is well written and accurately referenced, understandable, and readable.  I suggest its publication in the present form.

Author Response

Reviewer 3:

Thank you for the opportunity to review this interesting paper. The manuscript presented a comprehensive review of ATRA in APL and beyond. Indeed, outside the APL setting, the authors clearly reported synergistic differentiation effects when ATRA is combined with several other compounds, including BCL-2 inhibitors and epigenetic modulators, with the potential for these therapeutic associations to be tested in clinical trials for the treatment of non-APL AML. The manuscript is well written and accurately referenced, understandable, and readable.  I suggest its publication in the present form.

I sincerely thank the reviewer for the positive and thoughtful evaluation of my manuscript. I am pleased that the reviewer found the review comprehensive and clearly presented, and I appreciate the recognition of the therapeutic potential of ATRA-based combination strategies in non-APL AML. I am grateful for the reviewer’s supportive comments and recommendation for publication.

Round 2

Reviewer 1 Report

Comments and Suggestions for Authors

I appreciate the authors’ careful and largely constructive responses to the first-round comments. The revised manuscript is improved in clarity, balance, and transparency, particularly with respect to the literature search description, articulation of novelty, and acknowledgment of current clinical limitations. Overall, the manuscript is approaching an acceptable form. I have a small number of remaining comments that I believe would further strengthen its readability and translational value.

Major Comments

1. The authors note that the tables already include information on study type and trial phase, which is true. However, given the very broad spectrum of combinations covered—ranging from guideline-established APL regimens to early pre-clinical concepts—readers would still benefit from a more explicit, standardized indication of development stage.

I recommend adding a concise column such as “Status” (e.g., Approved/Guideline standard; Clinical trial; Early clinical; Pre-clinical) to the main clinical tables, and a “Model level” column (cell line / primary blasts / in vivo) to pre-clinical tables. This would substantially improve at-a-glance interpretation without altering the content.

2. I understand the authors’ rationale for not repeating detailed mechanistic figures from their previous reviews. That said, the current manuscript covers an unusually wide range of ATRA combination partners, and a high-level schematic (rather than a detailed pathway diagram) would greatly aid readers.

3. The revised Conclusion appropriately introduces the idea of integrating ATRA-based strategies with contemporary regimens. However, the discussion of lower-intensity therapy would benefit from clearer alignment with current standards, particularly HMA + venetoclax as the dominant backbone in older/unfit AML.

4. The added “Search strategy and study selection” paragraph is appropriate for a narrative review; optionally, the authors may note whether reference list screening or clinical trial registry searches were also considered.

5. The newly added paragraph summarizing clinical limitations is helpful; specifying that most non-APL data derive from small, non-randomized or exploratory studies would further strengthen the balance.

Author Response

Reviewer 1:

I appreciate the authors’ careful and largely constructive responses to the first-round comments. The revised manuscript is improved in clarity, balance, and transparency, particularly with respect to the literature search description, articulation of novelty, and acknowledgment of current clinical limitations. Overall, the manuscript is approaching an acceptable form. I have a small number of remaining comments that I believe would further strengthen its readability and translational value.

I sincerely thank Reviewer 1 for the careful and constructive evaluation of the revised manuscript. I appreciate the positive assessment of the improvements in clarity, balance, and transparency, and I am grateful for the additional suggestions, which have helped me further strengthen the readability and translational relevance of this review.

Major Comments

1. The authors note that the tables already include information on study type and trial phase, which is true. However, given the very broad spectrum of combinations covered—ranging from guideline-established APL regimens to early pre-clinical concepts—readers would still benefit from a more explicit, standardized indication of development stage. I recommend adding a concise column such as “Status” (e.g., Approved/Guideline standard; Clinical trial; Early clinical; Pre-clinical) to the main clinical tables, and a “Model level” column (cell line / primary blasts / in vivo) to pre-clinical tables. This would substantially improve at-a-glance interpretation without altering the content.

I agree with this suggestion and have revised the tables accordingly. To accommodate space limitations while improving clarity, I did not add new columns but instead standardized existing ones. In the clinical tables, the original “type of study/phase of trial” column has been revised to a unified “Status” category (Approved/Guideline standard; Clinical trial; Early clinical; Pre-clinical), with the study phase indicated within the same entry where applicable. Similarly, in the pre-clinical tables, the original “Cells examined” column has been revised to “Model level”, categorizing studies as cell line, primary blasts (specimens), or in vivo models, while retaining the specific cell or model names.

These revisions provide clearer at-a-glance information on developmental stage and experimental context, while preserving the original content and overall table structure.

2. I understand the authors’ rationale for not repeating detailed mechanistic figures from their previous reviews. That said, the current manuscript covers an unusually wide range of ATRA combination partners, and a high-level schematic (rather than a detailed pathway diagram) would greatly aid readers.

I appreciate this suggestion. In response, I note that the manuscript already includes Figure 1, which provides a high-level schematic overview of the ATRA-based combination strategies discussed in this review. This figure intentionally presents a conceptual outline rather than a detailed pathway diagram, in order to avoid redundancy with previously published mechanistic reviews. I believe that this schematic fulfills the intended purpose of offering readers an integrative and visual summary covered in this article.

3. The revised Conclusion appropriately introduces the idea of integrating ATRA-based strategies with contemporary regimens. However, the discussion of lower-intensity therapy would benefit from clearer alignment with current standards, particularly HMA + venetoclax as the dominant backbone in older/unfit AML.

I agree and have ensured that the “Conclusion and Future Directions” section explicitly aligns ATRA-based strategies with current treatment backbones. In particular, I now clearly frame lower-intensity approaches in the context of HMA + venetoclax–based regimens, highlighting their relevance for older or unfit AML patients and discussing how ATRA-based enhancers might be integrated into these standards. I believe this revision improves translational clarity and clinical relevance (page 29, line 804).

In contemporary AML practice, lower-intensity regimens for older or unfit patients are largely centered on hypomethylating agent (HMA) plus venetoclax combinations [207].

4. The added “Search strategy and study selection” paragraph is appropriate for a narrative review; optionally, the authors may note whether reference list screening or clinical trial registry searches were also considered.

I thank the reviewer for this suggestion. As this work is intended as a narrative review, the literature search was primarily conducted using PubMed, as described in the manuscript. Formal clinical trial registry searches were not performed. I have now clarified this point in the text to enhance transparency regarding the scope and methodology of study selection (page 3, line 91).

In addition, reference lists of relevant review articles were manually screened to identify further eligible studies; formal clinical trial registry searches were not performed.

5. The newly added paragraph summarizing clinical limitations is helpful; specifying that most non-APL data derive from small, non-randomized or exploratory studies would further strengthen the balance.

I agree with the reviewer and have further clarified this point in the Discussion by explicitly noting that, outside of APL, most evidence for ATRA-based combinations is derived from small, non-randomized, or exploratory studies (page 29, line 794).

Notably, outside of APL, most available data are derived from small, non-randomized, or exploratory studies, underscoring the need for prospective and well-controlled clinical trials.

I thank the reviewer again for these thoughtful and helpful comments, which have substantially strengthened the manuscript. I believe that the revisions made in response to these suggestions have improved both the clarity and the clinical relevance of the review.

Round 3

Reviewer 1 Report

Comments and Suggestions for Authors

I have reviewed the revised version of the manuscript carefully. Overall, the authors have responded appropriately to the previous round of comments, and the manuscript has improved in clarity, balance, and usability for readers.

Most of the substantive concerns raised earlier have now been satisfactorily addressed. In particular:

  • The scope and novelty of the review are now clearly articulated in the Introduction.

  • The description of the literature search and study selection is appropriate for a narrative review.

  • Clinical limitations of ATRA-based combination strategies, especially in non-APL AML, are now explicitly acknowledged.

  • The relative developmental maturity of different combination strategies is more clearly summarized in the Discussion and tables.

  • Language and formatting issues have been substantially improved compared with earlier versions.

At this stage, I have no major scientific concerns. I offer only a few minor, optional suggestions for final polishing:

Minor Suggestions

1. The current tables are now largely clear. If space permits, a brief legend note clarifying the meaning of “clinical,” “early-phase,” and “pre-clinical” designations (rather than adding new columns) could further help non-specialist readers, but this is not essential.

2. The revised conclusion appropriately mentions integration with current AML backbones. One concise sentence reinforcing that most ATRA-based combinations outside APL remain investigational and should be evaluated primarily within clinical trials would further emphasize a cautious and balanced clinical message.

3. A final light copyedit to catch any remaining spacing or hyphenation artifacts would be beneficial, but no substantive language issues remain.

Author Response

I have reviewed the revised version of the manuscript carefully. Overall, the authors have responded appropriately to the previous round of comments, and the manuscript has improved in clarity, balance, and usability for readers.

We sincerely thank the reviewer for the time and effort devoted to this careful evaluation.

Most of the substantive concerns raised earlier have now been satisfactorily addressed. In particular:

The scope and novelty of the review are now clearly articulated in the Introduction.

The description of the literature search and study selection is appropriate for a narrative review.

Clinical limitations of ATRA-based combination strategies, especially in non-APL AML, are now explicitly acknowledged.

The relative developmental maturity of different combination strategies is more clearly summarized in the Discussion and tables.

Language and formatting issues have been substantially improved compared with earlier versions.

At this stage, I have no major scientific concerns. I offer only a few minor, optional suggestions for final polishing:

Minor Suggestions

  1. The current tables are now largely clear. If space permits, a brief legend note clarifying the meaning of “clinical,” “early-phase,” and “pre-clinical” designations (rather than adding new columns) could further help non-specialist readers, but this is not essential.

I thank the reviewer for this helpful suggestion. In response, I have added a brief explanatory note to the Table 1 through Table 5 legend clarifying the meaning of the terms clinicalearly clinical, and pre-clinical. This addition is intended to facilitate interpretation by non-specialist readers, without adding new columns or altering the table structure.

  1. The revised conclusion appropriately mentions integration with current AML backbones. One concise sentence reinforcing that most ATRA-based combinations outside APL remain investigational and should be evaluated primarily within clinical trials would further emphasize a cautious and balanced clinical message.

I thank the reviewer for this valuable suggestion. In response, I have added the following sentence to the Conclusion to reinforce a cautious and balanced clinical message: “Outside of APL, however, most ATRA-based combination strategies remain investigational and should currently be evaluated primarily within the context of well-designed clinical trials.” (page 29, line 740).

  1. A final light copyedit to catch any remaining spacing or hyphenation artifacts would be beneficial, but no substantive language issues remain.

I thank the reviewer for this suggestion. I have carefully reviewed the manuscript and corrected remaining formatting inconsistencies where identified. While no substantive language changes were required, these edits were made to further improve overall readability and presentation.

Round 4

Reviewer 1 Report

Comments and Suggestions for Authors

I have reviewed the latest revised version of the manuscript. The authors have now satisfactorily addressed all major and minor points raised in previous rounds of review.

The scope and novelty of the review are clearly defined, the organization is logical and reader-friendly, and the balance between established clinical practice and emerging pre-clinical strategies is appropriate. Importantly, the manuscript now consistently distinguishes between approved regimens, early clinical evidence, and experimental approaches, and the clinical limitations of ATRA-based combination strategies—particularly outside APL—are clearly acknowledged.

The text is scientifically sound, carefully written, and well referenced. Only routine editorial polishing may be required. I have no further substantive comments and consider the manuscript suitable for publication.